# Spatially displaced excitation contributes to the encoding of interrupted motion by a retinal direction-selective circuit

Jennifer Ding[1,2], Albert Chen[3], Janet Chung[2], Hector Acaron Ledesma[4], Mofei Wu[2], David M Berson[5], Stephanie E Palmer[1,3,6]*, Wei Wei[1,2,6]*

[1]Committee on Neurobiology Graduate Program, The University of Chicago, Chicago, United States; [2]Department of Neurobiology, The University of Chicago, Chicago, United States; [3]Department of Organismal Biology, The University of Chicago, Chicago, United States; [4]Graduate Program in Biophysical Sciences, The University of Chicago, Chicago, United States; [5]Department of Neuroscience and Carney Institute for Brain Science, Brown University, Providence, United States; [6]Grossman Institute for Neuroscience, Quantitative Biology and Human Behavior, The University of Chicago, Chicago, United States

**Abstract** Spatially distributed excitation and inhibition collectively shape a visual neuron's receptive field (RF) properties. In the direction-selective circuit of the mammalian retina, the role of strong null-direction inhibition of On-Off direction-selective ganglion cells (On-Off DSGCs) on their direction selectivity is well-studied. However, how excitatory inputs influence the On-Off DSGC's visual response is underexplored. Here, we report that On-Off DSGCs have a spatially displaced glutamatergic receptive field along their horizontal preferred-null motion axes. This displaced receptive field contributes to DSGC null-direction spiking during interrupted motion trajectories. Theoretical analyses indicate that population responses during interrupted motion may help populations of On-Off DSGCs signal the spatial location of moving objects in complex, naturalistic visual environments. Our study highlights that the direction-selective circuit exploits separate sets of mechanisms under different stimulus conditions, and these mechanisms may help encode multiple visual features.

*For correspondence:
sepalmer@uchicago.edu (SEP);
weiw@uchicago.edu (WW)

## Introduction

How do sensory systems convert sensory inputs into behaviorally relevant neural signals? This question has been extensively investigated in the early visual system, where a neuron's responses to a set of parameterizable visual stimuli can be systematically probed to reveal a cell's receptive field (RF) properties in space and time. It has been increasingly appreciated that different visual stimuli can engage different mechanisms to shape the neuronal RF even at the earliest stage of visual processing in the retina. The transformation from the visual input to a retinal ganglion cell's spiking output is influenced by spatiotemporal patterns of stimuli in both the RF center and surround regions, highlighting the necessity of using diverse, ethologically relevant visual stimuli for delineating RF properties and for ultimately understanding neural coding of the animal's natural environment (*Chiao and Masland, 2003*; *Demb et al., 1999*; *Deny et al., 2017*; *Huang et al., 2019*; *Olveczky et al., 2003*; *Takeshita and Gollisch, 2014*; *Yao et al., 2018*).

Direction-selective ganglion cells (DSGCs) in the mammalian retina are well-studied for their motion direction selectivity. A DSGC fires maximally to visual stimuli moving across its RF in its preferred direction and is inhibited from firing by stimuli moving in the opposite, null direction (*Barlow and Hill, 1963*; *Barlow and Levick, 1965*; *Oyster, 1968*). The direction-selective spiking is

largely attributed to the GABAergic input from the starburst amacrine cell (SAC). SAC dendrites are inherently direction-selective, as they are activated by centrifugal motion, or motion from soma to dendritic tip (*Euler et al., 2002*). Additionally, only SAC dendrites that extend along the null direction of the DSGC selectively make GABAergic synapses with the DSGC (*Briggman et al., 2011*; *Fried et al., 2002*; *Lee et al., 2010*; *Wei et al., 2011*; *Yonehara et al., 2011*). Both the intrinsic properties of the SAC and the 'antiparallel' wiring patterns between the SAC and the DSGC are necessary for a strong null-direction inhibition onto the DSGC. The asymmetry of inhibition evoked by motion in the preferred and null directions is important for the DSGC's direction selectivity (*Pei et al., 2015*; *Taylor and Vaney, 2002*). The most well-studied DSGC type, the On-Off DSGC, prefers motion in one of the four cardinal directions (*Oyster and Barlow, 1967*; *Sabbah et al., 2017*). They have bistratified dendritic arbors in the On and Off sublamina of the inner plexiform layer (IPL) to extract motion directions of bright and dark signals, respectively (*Figure 1A*; *Famiglietti, 1983*; *He and Masland, 1998*; *Kittila and Massey, 1997*).

The RF of On-Off DSGCs has been studied with conventional visual stimuli such as stationary and moving spots, bars, and gratings. For motion stimuli that traverse across the entire RF, On-Off DSGC responses remain direction-selective over a broad range of contrast, luminance, speed, and background noise levels (*Barlow and Levick, 1965*; *Chen et al., 2016*; *Grzywacz and Amthor, 2007*; *Lipin et al., 2015*; *Sethuramanujam et al., 2016*; *Sivyer et al., 2010*; *Wyatt and Daw, 1975*). However, motion stimuli restricted to the distal RF subregion on the preferred side (defined as the side from which the preferred-direction moving stimulus approaches the RF) can elicit non-directional firing (*He et al., 1999*; *He and Masland, 1998*; *Rivlin-Etzion et al., 2011*; *Trenholm et al., 2011*). Based on the responses to moving stimuli presented to different subregions of the DSGC RF, the cell's RF structure can be viewed as consisting of multiple 'DS subunits' and a 'non-DS zone' at the edge of the preferred side. However, the neural mechanisms underlying the modular and heterogenous RF subunits of On-Off DSGCs have not been elucidated. Furthermore, the functional significance of this fine RF structure is not clear.

In this study, we investigated the spatial RF structure of the mouse On-Off DSGC subtype that prefers motion in the posterior direction of the visual field (pDSGC). We found that the pDSGC spiking RF is skewed toward the preferred side of the cell for both stationary and moving stimuli, even in the absence of SAC-mediated inhibition. Combining anatomical and functional analyses, we found a spatially non-uniform glutamatergic excitatory conductance that contributes to this spatial displacement. As a result of the displaced RF, moving stimuli that only activate the preferred side of the pDSGC RF trigger robust firing during both preferred and null direction motion. Theoretical analyses of the On-Off DSGC population response allow us to speculate about the ethological relevance of the displaced RF in processing complex natural scenes, suggesting that it can allow for better estimation of object location when a moving object emerges from behind an occluder. We term this type of motion 'interrupted motion' to distinguish it from more standard smooth motion stimuli. This phenomenon might also allow synchronous firing from different subtypes of DSGCs to serve as a useful alarm signal in complex scenes.

## Results

### Glutamatergic excitation of the On-Off pDSGC is spatially asymmetric relative to the soma

To investigate the spatial distribution of excitatory inputs to On-Off pDSGCs, we targeted pDSGCs in the Drd4-GFP transgenic mouse line. Bright stationary spots were presented to the periphery of the DSGC RF and the spiking responses were recorded. These spots were 110 μm in diameter and centered 165 μm from the soma of the recorded DSGC (*Figure 1B*, schematic). Measurement of the average radius of pDSGC On and Off dendrites (On: 88.1, STDEV 13.9 μm, Off: 75.9, STDEV 13.7 μm, from 25 cells, see Materials and methods) indicates that the spot mostly covered areas beyond the dendritic span of these cells. For each cell tested, we also presented a moving bar stimulus to confirm its directional tuning to posterior-direction motion in the visual field.

The spot stimulus delivered to the pDSGC RF periphery uncovered an asymmetric RF organization where some spots evoked maximal spiking (Max) and other spots presented to the opposite regions (Opp) across the dendritic span evoked minimal responses (*Figure 1B–C*, *Figure 1—figure*

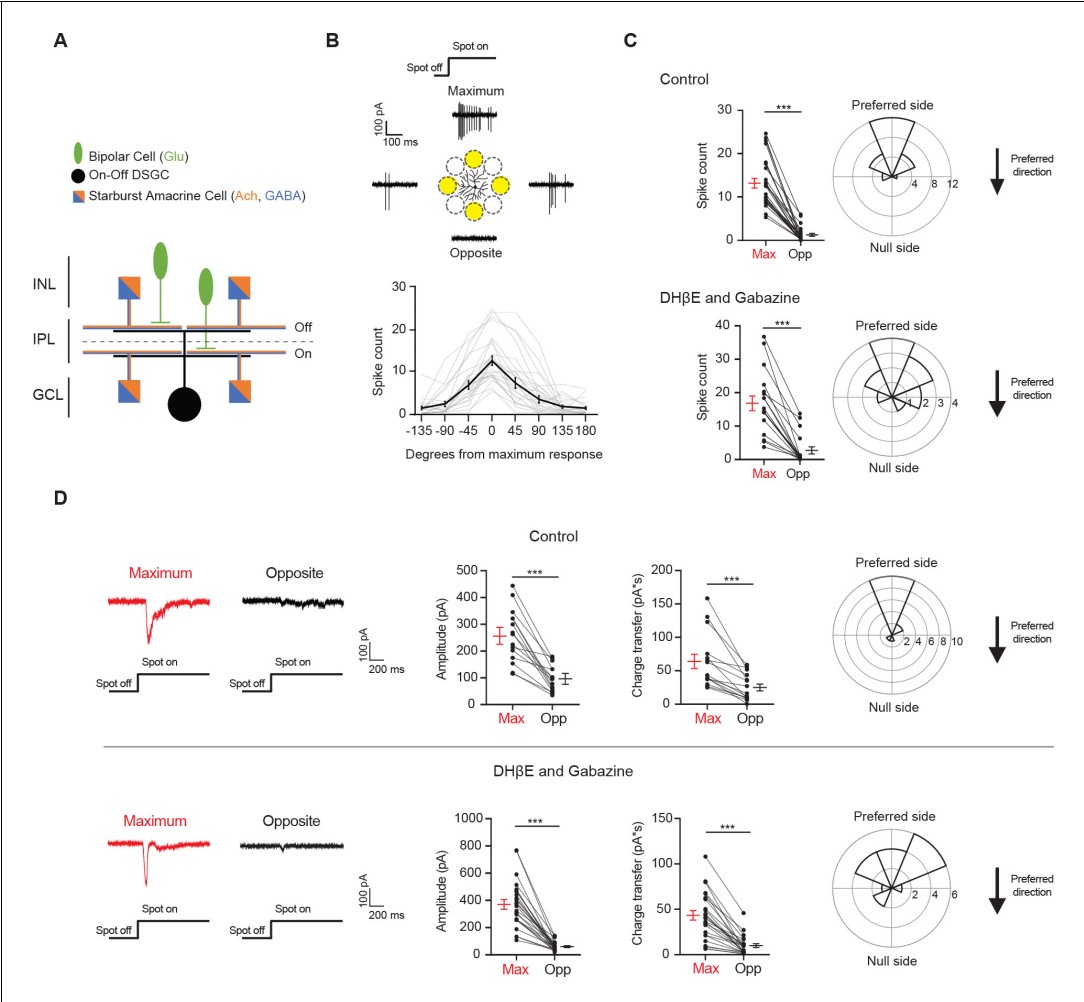

**Figure 1.** Drd4-GFP-labeled pDSGCs have spatially asymmetric glutamatergic receptive fields. (**A**) Schematic showing types of presynaptic neurons to an On-Off DSGC and the neurotransmitters they use. (**B**) Top: Example On spiking responses of a pDSGC to four different peripheral spots presented around dendritic span. Bottom: Individual (gray) and mean (black) pDSGC On spike counts evoked by spots presented at different locations (25 cells). (**C**) Top left: Pairwise comparison of mean spike counts in regions evoking the maximum number of On spikes (Max) and the opposite region (Opp) in the control condition (25 cells). Top right: Polar histogram of Max region locations aligned to the preferred-null motion axis. Radius indicates number of cells. Bottom: Same as in top but experiments performed in DHβE + Gabazine (18 cells). (**D**) Left: Example On EPSC responses to spots shown in the regions evoking the strongest EPSCs (Max) and the opposite region (Opp) in control (top) and in DHβE + Gabazine (bottom). Middle: Pairwise comparisons of EPSC amplitude and charge transfer to spots presented in the Max region and Opp region (Control: 15 cells; DHβE + Gabazine: 24 cells). Right: Polar histograms of Max region locations determined by EPSC charge transfer aligned to the preferred-null motion axis. Radius indicates number of cells. Summary statistics are mean ± SEM, ***p<0.001.

The online version of this article includes the following source data and figure supplement(s) for figure 1:

**Source data 1.** Drd4-GFP-labeled pDSGCs have spatially asymmetric glutamatergic receptive fields.

**Figure supplement 1.** pDSGCs have spatially asymmetric RFs.

**Figure supplement 1—source data 1.** pDSGCs have spatially asymmetric RFs.

supplement 1A-B). The Max-Opp axis largely corresponds to the preferred-null motion axis, which is shown by the polar plots in *Figure 1* where the preferred side is aligned to the top. The preferred side is the side of the receptive field that a bar moving in the cell's preferred direction enters first. The majority of the maximum responses to the peripheral spots occurs within 67.5 degrees from the preferred direction of the DSGC and half occur within 22.5 degrees of the preferred direction (*Figure 1B–C*, 'Control', *Figure 1—figure supplement 1A*).

Because a well-documented asymmetry in the direction-selective circuit is the asymmetric inhibition from SACs to DSGCs, we next tested whether the displacement of the pDSGC spiking RF is

eliminated by blocking SAC inputs. We perfused the retina with the nicotinic antagonist DHβE and the GABA$_A$ receptor antagonist gabazine to block these inputs. Under this condition, we still observed a spatial asymmetry in pDSGC spiking activity evoked by the flashing spots (*Figure 1C*, *Figure 1—figure supplement 1A-B*, 'DHβE and Gabazine'), indicating a spatial displacement of glutamatergic excitation of the pDSGC in the absence of SAC influence.

To directly measure the strength of excitatory inputs to pDSGCs at different stimulus locations, we performed whole-cell voltage clamp recordings of excitatory postsynaptic currents (EPSCs) evoked by peripheral flashing spots. Consistent with the pattern of pDSGC spiking activity, spot-evoked EPSCs show a spatial bias toward the preferred side (*Figure 1D*, *Figure 1—figure supplement 1C and F*, 'Control'). Isolation of the glutamatergic component of the EPSC by the addition of DHβE and gabazine confirms the persistence of the spatial asymmetry, indicating that the glutamatergic excitation of the pDSGC is not isotropic but is spatially displaced relative to the soma (*Figure 1D*, *Figure 1—figure supplement 1C-G*, 'DHβE and Gabazine'). We noted that gabazine and DHβE reduced the rise and decay time of the EPSC waveform compared to that in the control condition (*Figure 1—figure supplement 1E*), presumably due to the removal of the cholinergic component of the EPSC. Ablating the GABAergic contribution pharmacologically also rules out the possibility that a strong null-direction GABAergic inhibition is contaminating and artificially reducing the EPSCs on the null side, which is a potential confound during voltage clamp recordings due to imperfect control of membrane potential in distal dendrites (*Poleg-Polsky and Diamond, 2011*).

## Non-uniform excitatory conductance across the preferred-null motion axis contributes to the asymmetric glutamatergic RF

An asymmetric excitatory RF in a retinal neuron may result from asymmetric dendritic arbors and/or an asymmetric distribution of excitatory synaptic inputs across its dendritic field. We performed two-photon imaging of dye-filled dendritic arbors after recording glutamatergic EPSCs evoked by the peripheral flashing spot stimulus described above (*Figure 2A–B*). Consistent with previous studies, dendritic arbors of pDSGCs do not exhibit a salient or consistent bias relative to the cell's preferred motion direction. The total dendritic length or the number of dendritic branching points does not significantly differ between the preferred and null sides of the pDSGC dendritic field (*Figure 2C–D*, *Figure 2—figure supplement 1A-C*). This apparent randomness of dendritic arbor distribution relative to the cell's preferred motion direction contrasts with the previously reported mouse On-Off DSGC subtype preferring motion in the superior direction that have dendritic arbors strongly displaced to the null side of the soma (*El-Quessny et al., 2020*; *Kay et al., 2011*; *Trenholm et al., 2011*).

Presentations of peripheral spots revealed that pDSGC glutamatergic RFs are consistently skewed toward the preferred side (*Figure 1C–D*), while pDSGC dendritic fields are not (*Figure 2—figure supplement 1A-C*). This indicates that the functional glutamatergic RF of a pDSGC cannot be solely explained by its dendritic morphology. However, since pDSGC dendrites define the physical locations of the cell's excitatory postsynaptic sites, the spatial distribution of dendritic arbors may partially influence the position of the cell's excitatory RF. Indeed, we found a non-random skew of the pDSGC dendritic length and number of branch points toward the regions corresponding to light spots evoking the strongest glutamatergic EPSCs (*Figure 2E*). This indicates that the pDSGC's excitatory RF is partially shaped by its dendritic morphology, rather than being completely independent of the cell's dendritic distribution.

However, dendritic morphology cannot fully account for the pattern of glutamatergic excitatory RF displacement. When we examined the relationship between EPSC bias and dendritic arbor distribution along the axis of maximal glutamatergic displacement, we did not find a positive correlation between the extent of EPSC bias and the extent of dendritic bias (*Figure 2F*), suggesting that mechanisms other than dendritic arbor density also contribute to the displacement of the pDSGC glutamatergic RF.

To further explore the relationship between the pDSGC glutamatergic RF and its dendritic distribution, we performed another set of experiments to obtain a more complete RF map. First, the preferred direction of each cell's spiking activity was determined by loose cell-attached recordings using a moving bar stimulus with no synaptic blockers. Next, peripheral flashing spots were used to estimate the spatial displacement of the pDSGC glutamatergic RF in the presence of nicotinic and GABAergic receptor antagonists (DHβE + Gabazine) as described above. Then, a smaller, 20

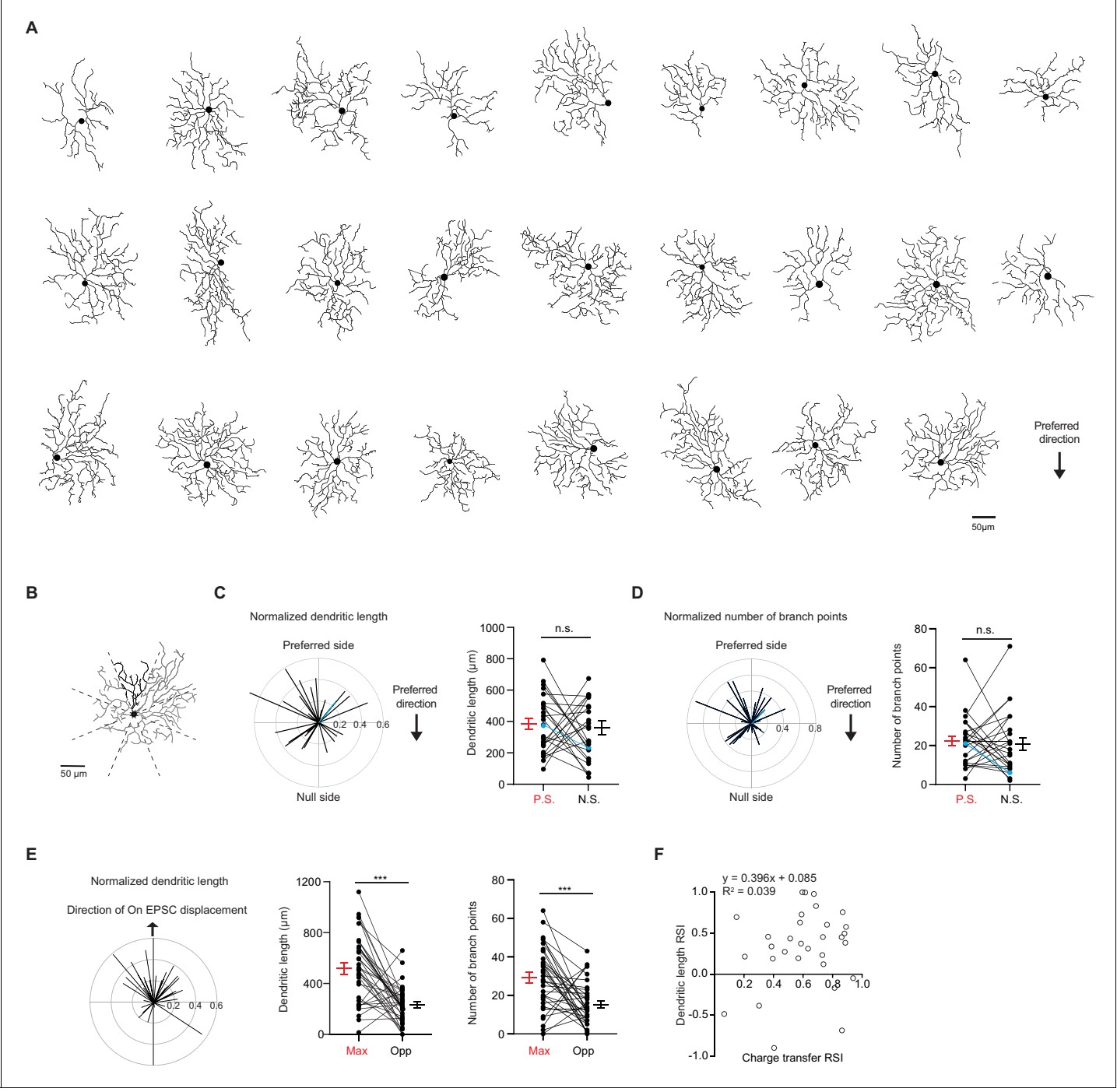

**Figure 2.** pDSGC dendritic morphology does not show a spatial bias toward the preferred side. (A) Traced On layer dendritic morphologies of pDSGCs aligned to their preferred-direction motion. (B) Example On morphology of a pDSGC cell divided into eight sectors for calculating normalized dendritic length vector and pairwise comparisons. (C) Left: Normalized vector sum of On dendritic length aligned to pDSGCs' preferred direction motion. Right: Pairwise comparison of dendritic length on the preferred vs null sides of each cell (26 cells, p=0.70). Blue represents example cell in **B**. (D) Left: Normalized vector sum of On branch points aligned to pDSGCs' preferred direction motion. Right: Pairwise comparison of branch points on the preferred vs null side (26 cells, p=0.68). Blue represents example cell in **B**. (E) Left: Normalized vector sum plot of On dendritic length in eight sectors aligned to the region evoking maximal glutamatergic EPSC (upward arrow). Middle: Total dendritic length in the region evoking maximal glutamatergic EPSC as determined by peripheral spot stimulus (Max) and opposite region (Opp) (31 cells, ***p<0.001). Right: Total number of branch points in the Max and Opp regions (31 cells, ***p<0.001). (F) On dendritic length RSI versus On EPSC charge transfer RSI (31 cells). RSI is based on regions evoking maximal glutamatergic charge transfer responses and opposite regions. Summary statistics are mean ± SEM.

The online version of this article includes the following source data and figure supplement(s) for figure 2:

*Figure 2 continued on next page*

*Figure 2 continued*

**Source data 1.** pDSGC dendritic morphology does not show a spatial bias toward the preferred side.
**Figure supplement 1.** Off dendritic morphology shows spatial bias along glutamatergic receptive field but not global preferred-null motion axis.
**Figure supplement 1—source data 1.** Off dendritic morphology shows spatial bias along glutamatergic receptive field but not global preferred-null motion axis.

µm diameter bright stationary spot was repeatedly flashed at random locations within a 11-by-11 220 µm square grid centered on the pDSGC soma. A heatmap of glutamatergic EPSC charge transfer evoked by the small flashing spot was generated and overlaid with the reconstructed dendritic arbors for each pDSGC (*Figure 3A*).

RF mapping with small flashing spots revealed a spatial displacement of glutamatergic EPSC distribution that aligns well with the displacement determined by larger spots presented to the pDSGC periphery (*Figure 3B*). Similar to the displacement pattern revealed by stimulation of the RF periphery, the strongest glutamatergic EPSCs are preferentially located on the preferred side of pDSGC

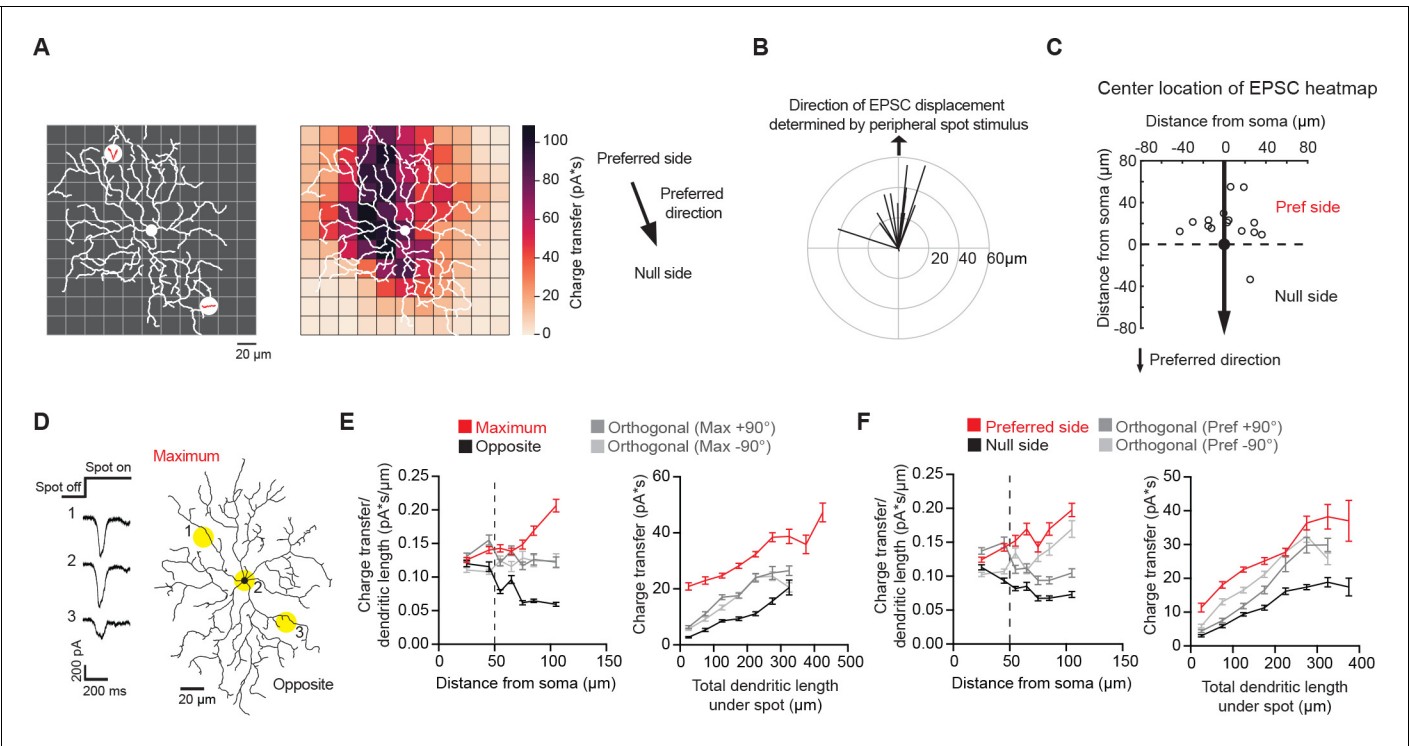

**Figure 3.** Non-uniform glutamatergic synaptic excitation across pDSGC dendritic span contributes to skewed excitatory receptive field. (**A**) Left: Schematic for small spot RF mapping experiment in DHβE + Gabazine. Middle: Example heat map. Right: Preferred motion direction of the example cell in relation to the preferred side and the null side of the cell's RF. (**B**) Vector sum plot of On EPSC charge transfer center of mass determined by small spot RF mapping aligned to the region where a peripheral spot evokes the maximum glutamatergic EPSC (upward arrow). (**C**) Spatial locations showing centers of mass of glutamatergic excitatory On charge transfer obtained from experiments illustrated in **A** aligned to each cell's preferred motion direction (15 cells). (**D**) Example On EPSC responses to spots presented to a pDSGC along the maximum-opposite axis of glutamatergic RF displacement. (**E**) Left: On 'EPSC current density' (i.e. ratio of charge transfer per dendritic length) versus distance from soma along the maximum-opposite axis of glutamatergic RF displacement (16 cells, ***p<0.001) as well as along the orthogonal axis. Right: Summary plot of glutamatergic charge transfer as a function of total dendritic length centered around the flashing spot for spots shown more than 50 µm away from the soma (16 cells, ***p<0.001). (**F**) Same as **E** but along the preferred-null motion axis (15 cells, charge transfer/dendritic length vs distance from soma **p=0.0026, charge transfer vs dendritic length ***p<0.001). Summary statistics are mean ± SEM.

The online version of this article includes the following source data and figure supplement(s) for figure 3:

**Source data 1.** Non-uniform glutamatergic synaptic excitation across pDSGC dendritic span contributes to skewed excitatory receptive field.
**Figure supplement 1.** pDSGC glutamatergic synaptic excitation is displaced relative to the dendritic field.
**Figure supplement 1—source data 1.** pDSGC glutamatergic synaptic excitation is displaced relative to the dendritic field.

somas (*Figure 3C*). Notably, for most cells, the center of the glutamatergic EPSC RF is displaced from the center of the dendritic field (*Figure 3—figure supplement 1A-B*), indicating additional mechanisms underlying the glutamatergic RF displacement apart from the dendritic arbor distribution.

We further examined the strength of the glutamatergic input across the pDSGC dendritic field (*Figure 3D*). We normalized the EPSC charge transfer by the total dendritic length in a circle with a 60 µm diameter centered on each small flashing spot. This provides an estimate of the strength of glutamatergic inputs per unit dendritic length, or 'EPSC density', at each stimulus location. When we calculated the density along the axis of maximal glutamatergic RF displacement, we found that the EPSC density on the displaced side of the dendritic field is larger than the corresponding region on the opposite side (*Figure 3E*, left, *Figure 3—figure supplement 1E*). From 50 µm away from the soma, spots on the side corresponding to the maximum displacement (Maximum) yield stronger EPSCs than spots on the opposite side (Opposite), while controlling for the same dendritic length (*Figure 3E*, right, *Figure 3—figure supplement 1E*). This heterogeneity in EPSC density across the dendritic span persists when comparing EPSC responses along the preferred-direction motion axis of the cell (*Figure 3F*, *Figure 3—figure supplement 1F*). Analysis of the peak amplitudes showed similar results as charge transfer (*Figure 3—figure supplement 1C-D*).

Next, we asked what mechanisms could underlie the non-uniform excitatory conductance across the dendritic span. Because bipolar cells provide the dominant excitatory input to DSGCs, the enhanced excitation on the preferred side of the receptive field could stem from increased density or strength of bipolar synapses. Using a published serial block-face scanning electron microscopic (SBEM) dataset of the adult mouse retina (*Ding et al., 2016*), we reconstructed large numbers of On starburst amacrine cells and On-Off DSGCs (*Figure 4A*). The preferred direction of the traced DSGCs was inferred from the mean orientation of starburst dendrites making 'wraparound' synapses onto the DSGCs, following the previous work of *Briggman et al., 2011* (*Figure 4C*). Among all DSGCs, four preferred directions were seen roughly 90 degrees apart, as expected for most retinal locations (*Sabbah et al., 2017*). We selected three On-Off DSGCs for which the inferred preferred direction of motion fell on the horizontal axis like the cells we studied electrophysiologically, two with one motion preference, and the other with the opposite preference. Technical constraints prevented us from saying which cells preferred posterior motion and which cells preferred anterior motion (see Materials and methods). We then mapped the distribution of bipolar ribbon synapses onto the dendrites of these three cells. We found no marked gradient in ribbon density across the preferred-null motion axis of the three cells (*Figure 4D–F*, *Figure 4—figure supplement 1A-B*), suggesting that the gradient in EPSC density across this axis is not determined by the density of bipolar inputs.

In addition to ribbon density, we also examined the time course of glutamatergic EPSCs in the presence of DHβE and gabazine along the axis of maximal glutamatergic RF displacement and along the preferred-null motion axis. We found no significant differences in the latency, rise and decay times in EPSC waveforms across the axis of maximal glutamatergic RF displacement in response to the small flashing spots (*Figure 4G–I*, left panels). In addition, we did not detect a monotonic change in glutamatergic EPSC kinetics along the preferred-null motion axis in DHβE and gabazine (*Figure 4G–I*, right panels) or in DHβE only (*Figure 4—figure supplement 1C-D*). That we found no differences in the kinetics of EPSC waveforms across the pDSGC dendritic field contrasts with a previous study of On DSGCs reporting a gradient of EPSC kinetics from slow/sustained to fast/transient along the preferred-null motion axis (*Matsumoto et al., 2019*). This gradient in On DSGC EPSC kinetics is thought to arise from different bipolar cell subtypes, and may implement a Hassenstein-Reichardt-Detector-like mechanism for the On DSGCs' direction selectivity. Thus, our data suggest that the kinetics of bipolar cell signals onto On-Off pDSGCs do not differ substantially along the preferred-null motion axis.

## Null-direction response emerges during partial activation of the displaced pDSGC RF

Because On-Off DSGCs are thought to be dedicated to encoding object motion, we next asked how the displaced excitatory RF of the pDSGC revealed by the stationary flashing stimuli contributes to motion processing. First, we considered whether a displaced excitatory RF could benefit the direction-selective mechanisms during full-field smooth motion. By comparing the onset times of EPSCs

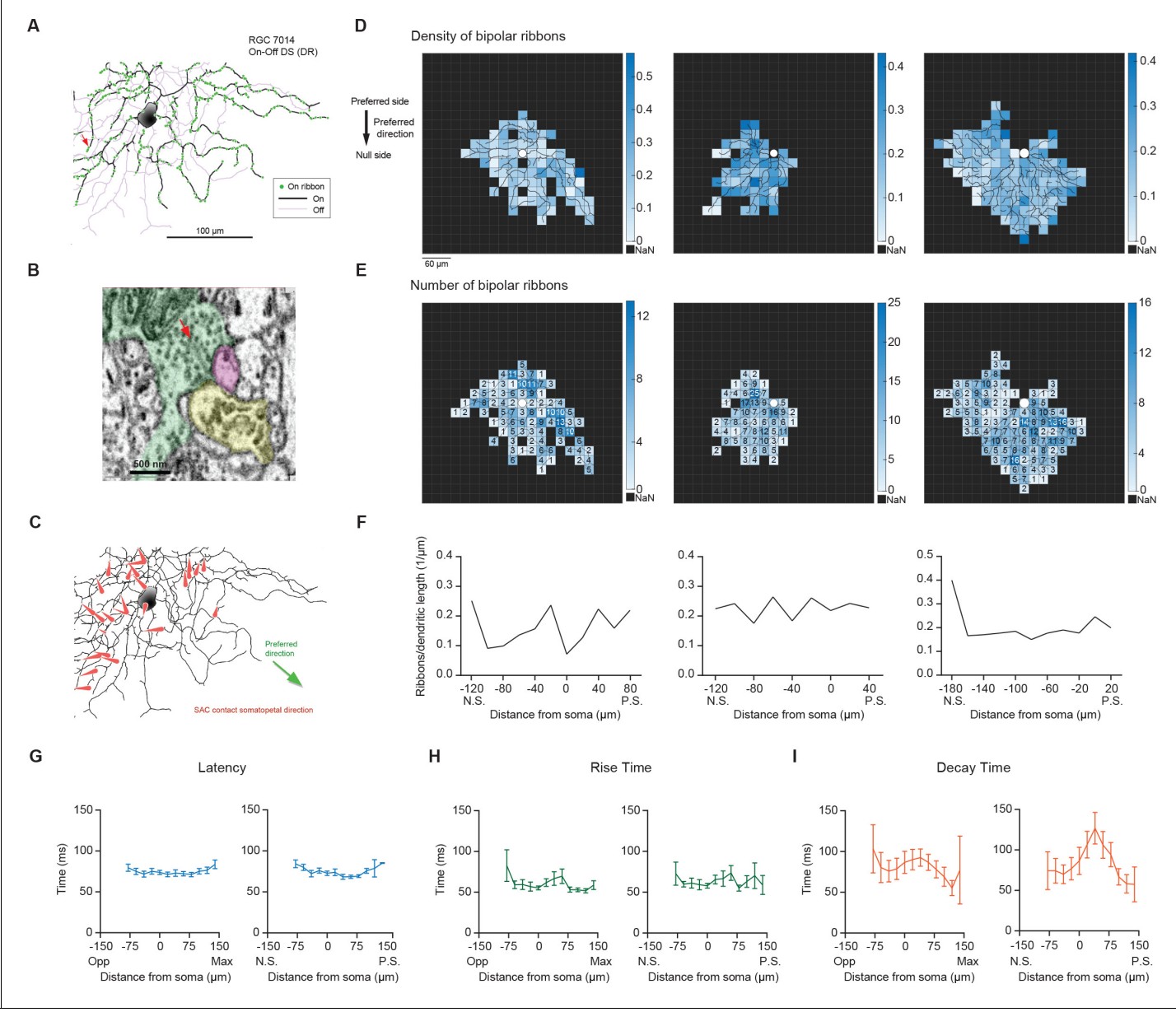

**Figure 4.** Neither the density of bipolar ribbon inputs nor the kinetics of glutamatergic EPSCs varies systematically along the axis of RF displacement. (A) Top: Distribution of ribbon synaptic input to one of the three reconstructed On-Off DSGCs. (B) Ultrastructure of the synapse indicated by the red arrow in **A**, as visualized by serial block-face electron microscopy. The presynaptic bipolar cell (green) belongs to Type 5t. Red arrow marks the ribbon. The magenta profile belongs to the On-Off DSGC. The other postsynaptic partner at this dyad synapse was another ganglion cell (yellow). Although only a fragment was included in the volume, it was presumably a DSGC, since it costratified entirely with the inner dendrites of the On-Off DSGC (**A**; black dendrites). (C) SAC inputs onto example On-Off DSGC. Red arrows indicate the direction and location of the SAC inputs on the DSGC dendrites. (D) Density map of bipolar ribbon synapses for three example On-Off DSGCs with estimated preferred directions along the posterior-anterior axis (see Methods). The soma location is indicated by the white spot in the center. (E) Bipolar ribbon heat map for the three example cells, respectively. The number of ribbons in each square are indicated. (F) Quantification of ribbon density across the preferred-null axis, respectively. The soma location is at 0. (N.S. = null side, P.S. = preferred side). (G) Left: Summary of latency of glutamatergic EPSC responses along the maximum-opposite axis (Max = maximum glutamatergic EPSC region, Opp = opposite region) (16 cells, p=0.75). Right: Summary of latency along the preferred-null motion axis (N.S. = null side, P.S. = preferred side)(15 cells, p=0.88). (H) Same as in **G**, except for rise time (10%–90%). (Left: p=0.12, right: p=0.21). (I) Same as in **G**, except for decay time (90%–30%). (Left: p=0.25, right: p=0.26). Summary statistics are mean ± SEM.

The online version of this article includes the following source data and figure supplement(s) for figure 4:

**Source data 1.** Neither the density of bipolar ribbon inputs nor the kinetics of glutamatergic EPSCs varies systematically along the axis of RF displacement.

*Figure 4 continued on next page*

*Figure 4 continued*

**Figure supplement 1.** Bipolar ribbon density does not consistently change across the preferred-null motion axis.

**Figure supplement 1—source data 1.** Bipolar ribbon density does not consistently change across the preferred-null motion axis.

and IPSCs to preferred-direction motion, we found that cells with more spatially separated excitatory and inhibitory receptive fields were more direction-selective (*Figure 7—figure supplement 1L*).

When motion trajectories are more complex, as often occurs in the natural environment, the displaced excitatory pDSGC RF may confer additional characteristics to the cell's motion encoding. We reasoned that because full-field smooth motion stimuli traverse the entire RF of the pDSGC, SAC-mediated inhibition exerts a dominant influence on pDSGC spiking by strongly suppressing the pDSGC null-direction response. However, when moving objects pass behind occluders or start moving from behind other objects, the RFs of certain DSGCs are partially activated, changing the interaction between SAC-mediated inhibition and displaced excitation. Therefore, we next investigated how the direction-selective circuit would process interrupted motion.

To examine if the displaced excitatory RF of the pDSGC plays a role in encoding interrupted motion, we created an occluded motion stimulus where a moving bar disappeared behind a central occluder 220 µm in diameter (*Figure 5A*). The occluder covered the dendritic span of the pDSGC and a substantial portion of its RF center. In contrast to the full-field motion stimulus, the occluded motion stimulus caused both preferred and null-direction spiking responses (*Figure 5B*, *Figure 5—figure supplement 1A-C*). These responses are only evoked when the bar travels in the preferred side, which corresponds to the displaced side of the pDSGC RF (*Figure 5B*, middle row, and **5C**). In contrast, no spiking response is evoked when the bar moves across to the null side beyond the occluder (*Figure 5B*, bottom row, *Figure 5C*). To test whether this regional difference is due to the asymmetric wiring between SACs and DSGCs, we blocked the cholinergic and GABAergic transmission with DHβE and gabazine and saw that the regional difference persisted (*Figure 5C*, *Figure 5—figure supplement 1B*). Consistent with the spiking pattern, the EPSC responses also reflected this regional asymmetry (*Figure 5D*). In contrast, IPSC responses to the occluded bar stimulus are displaced to the opposite side compared to EPSC responses (*Figure 5—figure supplement 1D*), consistent with the asymmetric wiring pattern from SACs from the null side to the DSGC (*Briggman et al., 2011*; *Fried et al., 2002*; *Lee et al., 2010*; *Wei et al., 2011*; *Yonehara et al., 2011*).

Moreover, we individually blocked each of several types of signaling, including $GABA_C$ receptor activity, glycine receptor activity, and gap-junction coupling with TPMPA, strychnine, MFA, and Carbenoxolone, respectively, and found that the spatial asymmetry of the spike response remains during the occluded motion stimulus (*Figure 5—figure supplement 1H-K*). Based on these pharmacology results, we conclude that the asymmetric glutamatergic RF of the pDSGC contributes to robust spiking in both null and preferred directions when the preferred side of the RF is activated by the occluded motion stimulus.

Since there are two On-Off DSGC subtypes that are tuned to opposite directions along the posterior-anterior axis, we investigated if the anterior-direction-selective DSGC (aDSGC) exhibits the same null-direction response pattern to the occluded motion stimulus as the pDSGC. To identify both DSGC subtypes, we performed calcium imaging of GCaMP6-expressing RGCs in another transgenic mouse line carrying Vglut2-IRES-Cre and floxed GCaMP6f (*Figure 5—figure supplement 2A*) during the full-field moving bar stimulus. We then used an online analysis to identify aDSGCs and pDSGCs based on the directional tuning of their calcium signals to the anterior and posterior motion directions (*Figure 5F*, *Figure 5—figure supplement 2B-C*). Next, we centered the occluded motion stimulus on individual aDSGC and pDSGC somas and performed calcium imaging during the occluded motion stimulus. Consistent with pDSGC spiking activity, a significant null-direction calcium response of the pDSGCs was evoked when the occluded moving bar traveled across the receptive field to the preferred side of the RF (*Figure 5G*, *Figure 5—figure supplement 2D*). Notably, aDSGCs also exhibited a null-direction response to the occluded motion stimulus that is similar to that of pDSGCs (*Figure 5H*, *Figure 5—figure supplement 2D*), indicating that aDSGC RFs are also displaced to the preferred side.

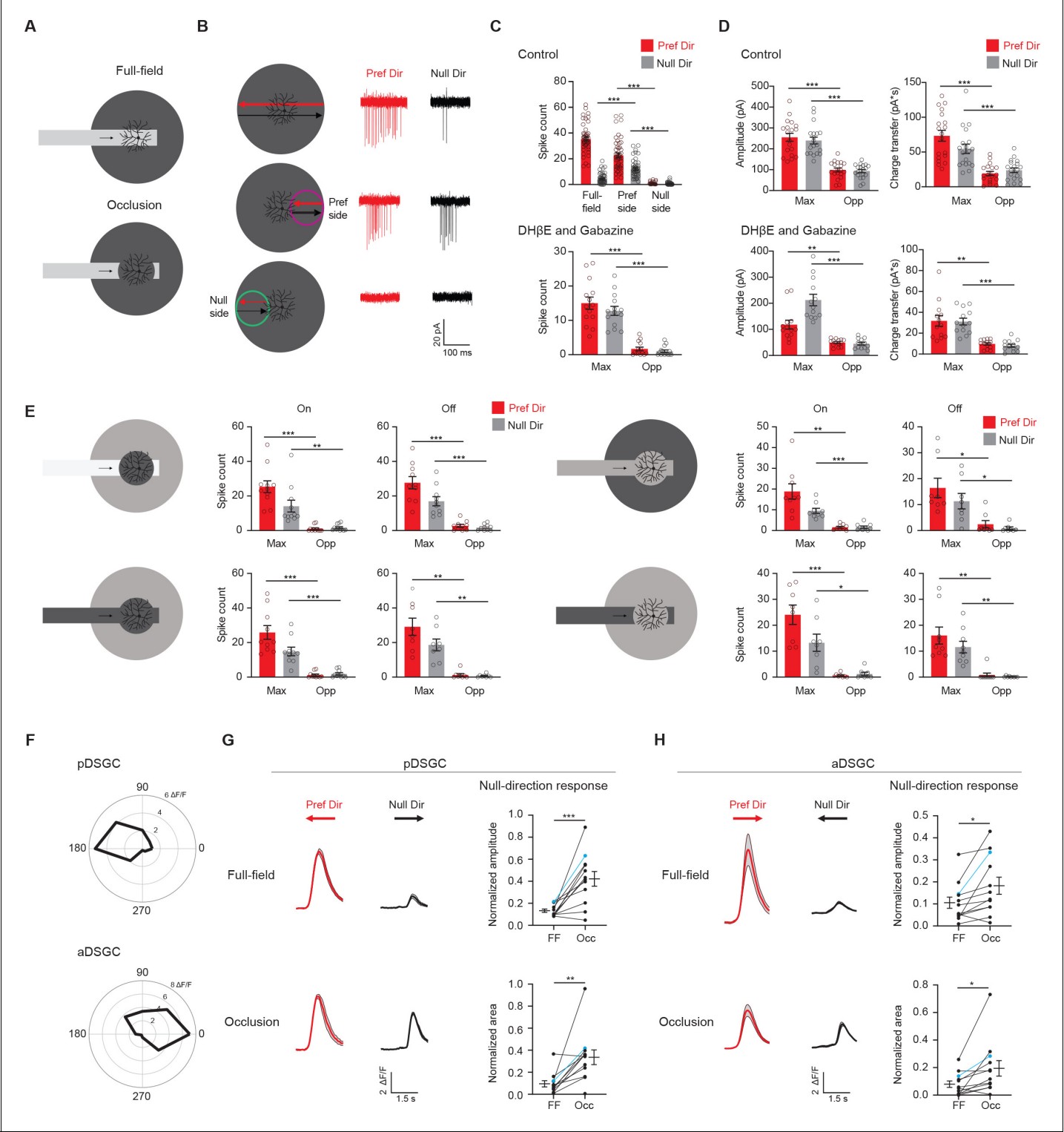

**Figure 5.** Displaced excitatory receptive field contributes to null-direction responses in the preferred region. (**A**) Full-field moving bar and occluded bar stimuli. (**B**) Example On spiking responses of a pDSGC to full-field bar (top) and occluded bar stimulus (middle, bottom) moving in the preferred (red) and null (black) directions. (**C**) Top: Mean spike counts of pDSGCs to the full-field moving bar (full-field) and the occluded moving bar on the preferred side and null side (48 cells). Bottom: Mean spike counts in DHβE + Gabazine to the occluded bar stimulus in the region evoking the maximum spiking (Max) and the opposite region (Opp) (14 cells). (**D**) Mean pDSGC EPSC peak amplitude and charge transfer in the region evoking the maximum response (Max) and the opposite region (Opp) in the control (top, 19 cells) and DHβE + Gabazine (bottom, 13 cells) conditions. (In DHβE + Gabazine: Pref Dir – Max vs Pref Dir – Opp amplitude **p=0.0028, Pref Dir – Max vs Pref Dir – Opp charge transfer **p=0.0012). (**E**) Mean On

*Figure 5 continued on next page*

*Figure 5 continued*

and Off spiking responses to occluded bar stimuli at different contrast configurations. Top left: On: 11 cells, Max null dir. Vs Opp null dir. **p=0.008. Off: 10 cells. Bottom left: On: 10 cells. Off: 8 cells. Top right: On: 9 cells, Max pref. dir. Vs. Opp pref. dir. **p=0.003. Off: 8 cells, Max pref. dir. Vs Opp. Pref. dir. *p=0.034, Max null dir. Vs Opp null dir. *p=0.02. Bottom right: On: 8 cells, Max null dir vs Opp null dir. *p=0.014. Off: 9 cells, Max pref. dir. Vs Opp pref. dir. **p=0.004, Max null dir. Vs Opp null dir. **p=0.002. (F) Top: Example directional tuning curve of GCaMP6 signal of a pDSGC. Bottom: Example tuning curve of an aDSGC. (G) Top left: Example GCaMP6 fluorescence traces of a pDSGC for the full-field moving bar in the preferred (red) and null (black) directions. Bottom left: Example GCaMP6 traces of the cell for the occluded bar stimulus. Shaded areas represent SEM. Top right: Normalized amplitude of the posterior-preferring cell null-direction response during the full-field bar and the occluded bar (12 cells). Bottom right: Normalized area of the null-direction response during the full-field bar and the occluded bar (12 cells, **p=0.0035). (H) Same as in G, except for aDSGCs (12 cells, normalized amplitude *p=0.013, normalized area *p=0.019). Summary statistics are mean ± SEM, ***p<0.001 except where specified otherwise.

The online version of this article includes the following source data and figure supplement(s) for figure 5:

**Source data 1.** Displaced excitatory receptive field contributes to null-direction responses in the preferred region.

**Figure supplement 1.** Displaced glutamatergic excitation contributes to null-direction responses in the preferred region.

**Figure supplement 1—source data 1.** Displaced glutamatergic excitation contributes to null-direction responses in the preferred region.

**Figure supplement 2.** Calcium imaging of anterior and posterior-preferring On-Off DSGCs.

**Figure supplement 2—source data 1.** Calcium imaging of anterior and posterior-preferring On-Off DSGCs.

The above results show that continuous motion interrupted by a stationary occluder in the center of the pDSGC's RF causes unexpected null-direction spiking as the bar emerges from behind the occluder into the preferred side of the pDSGC RF. However, this occluded motion stimulus cannot distinguish whether the interruption itself or only the start position of the emerging bar was necessary for the null-direction response. Therefore, we created a stimulus where a bar emerges at different locations along the preferred-null motion axis of the cell. The start motion of the bar activates

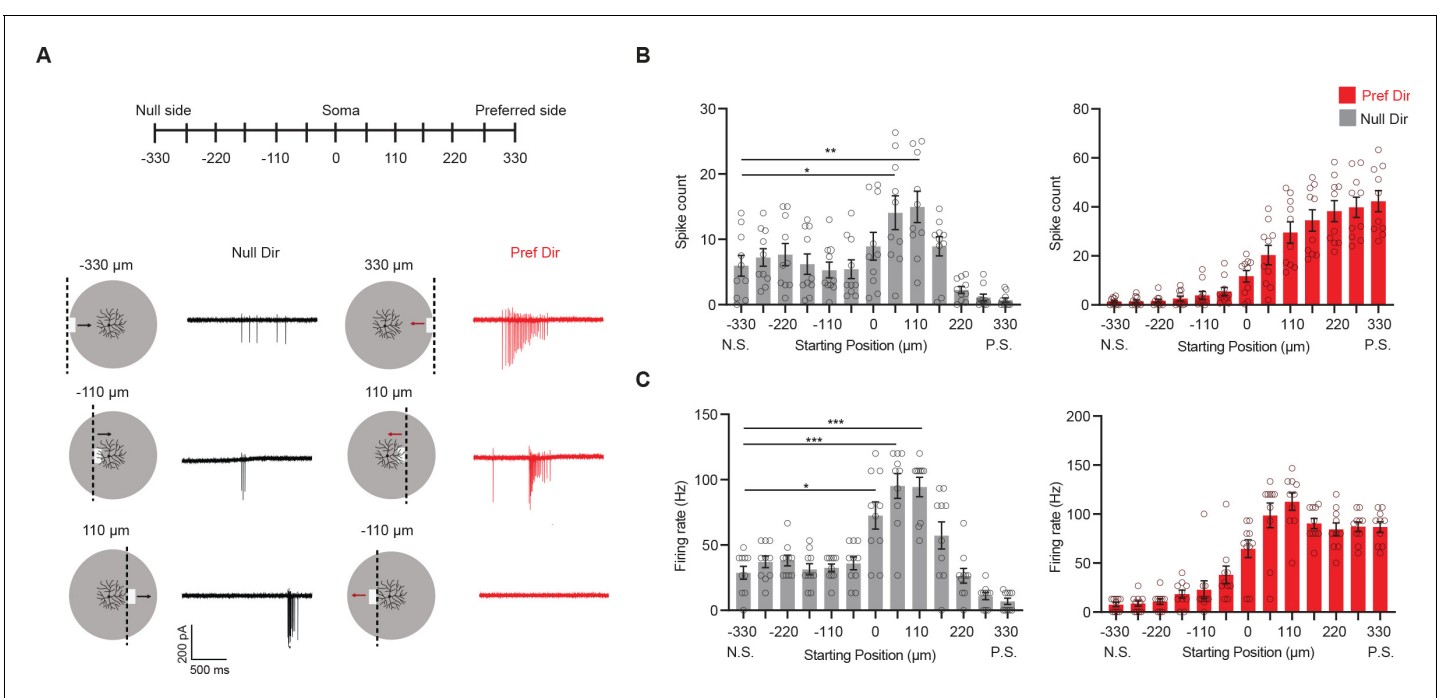

**Figure 6.** Null-direction response is dependent on start position of emerging bar. (A) Schematic of moving bars emerging from different locations along the DSGC's preferred-null motion axis and example spiking responses. The soma location is at 0. Vertical dashed lines on the schematic indicate the positions of the emerging leading edge of the moving bar. (B) Mean spike counts (null direction: −330 vs 55 *p=0.012, −330 vs 110 **p=0.0088) and (C) firing rates (null direction: −330 vs 0 *p=0.014) to bars emerging from different locations along the preferred-null motion axis (10 cells). Summary statistics are mean ± SEM, ***p<0.001 except where specified otherwise.

The online version of this article includes the following source data for figure 6:

**Source data 1.** Null-direction response is dependent on start position of emerging bar.

different parts of the DSGC's receptive field (*Figure 6A*). For the null-direction moving bar, there is an increase in both the spike number and the firing rate as the starting position of the moving bar is located past the soma on the preferred side of the pDSGC RF (*Figure 6A–C*). Thus, the emergent growing edge caused null-direction spiking of pDSGCs in a similar pattern as the moving bar emerging behind the central occluder. These results illustrate that the null-direction response of the pDSGC during the occluded motion stimulus is dependent on the position in the receptive field from which the moving edge emerges, not the previous motion approaching the occluder.

## Null-direction responses of DSGCs during partial activation of their RFs can be useful for decoding object location

We explored how the displaced RFs could functionally benefit the encoding of interrupted motion by DSGCs. Our experimental results show that pDSGCs generate robust spiking activity in both the preferred and null directions when a moving object appears and starts moving on the preferred side of the cell's RF, as when an object emerges from behind an occluder positioned over the cell's soma. We asked whether this prima facie aberrant signaling can have a functional role relevant to the behavioral goals of the organism.

We hypothesize that partial, non-directional RF activation of a DSGC may provide precise information about the spatial position of motion initiation. In particular, if a moving object emerging from behind an occluder activates the preferred side of an On-Off DSGC's RF, this DSGC would generate a null-direction spiking response together with the preferred-direction response of a nearby On-Off DSGC subtype preferring the opposite motion direction (*Figure 7A*). Such synchronous spiking activity between DSGCs of opposite preferred directions could yield a stronger and more localized spatial signal at the population level at the location of motion interruption.

To test how the null-direction response during interrupted motion might enhance downstream estimation of stimulus position, we simulated the spiking responses of pDSGCs in a population model consisting of On-Off DSGC subtypes that prefer opposite motion directions on the posterior-anterior axis (aDSGCs and pDSGCs, *Figure 7A*). One thousand cells of each subtype were arranged in a two-dimensional array with biologically realistic positional jitter, such that the spatial positions of the two subtypes were uncorrelated. We simulated a bright edge moving along the posterior-anterior axis in a single direction at a constant velocity under both uninterrupted and interrupted motion conditions and analyzed the On response to the edge. The mean spiking response of each DSGC was modeled as a rectified sine wave with parameters obtained from our experimental data, and sub-Poisson trial-to-trial variability was introduced to the mean spiking response on each trial (*Figure 7—figure supplement 1*). The spatial positions of the DSGCs were shuffled in each simulation block to allow us to sample different spatial arrangements of the RFs.

We used a labeled-line decoder (equivalent to a 'population vector' decoder) that estimated the spatial position of the moving bar edge as a weighted average of the RF center positions where the weights were determined by the firing rate (response strength) and RF width (response precision) of each cell (see Materials and methods). Using this decoder, we compared the scenario in which none of the DSGCs were occluded (full-field) with the scenario in which a single occluder was placed near the center of the population. We chose to implement the model with different low baseline firing rates to more faithfully represent biological noise. Previous reports show baseline firing rates up to 0.1 spike/s (*Yao et al., 2018*) for On-Off DSGCs. Our experiments yielded background noise levels more on the scale of 0–0.025 spike/s. Thus, we evaluated the computational model at three different low noise levels (*Figure 7D–E*).

Our modeling results show that when the moving bar emerges from behind the occluder, null-direction responses from the pDSGCs undergoing partial RF activation on the preferred side transiently degrades the population estimate of the motion direction as would be expected (*Figure 7C and E*). However, the coincidence of the null-direction responses from pDSGCs and the preferred-direction response of neighboring aDSGCs when the bar exits the occlusion substantially improves the estimation of bar position at this time (*Figure 7B and D*). For a bar traveling at the speed of 330 μm/s (the lowest speed in our experiments and simulations), the synchronous firing between pDSGCs and the neighboring aDSGCs reduces the error in the population estimate of the bar edge's spatial position by over 80 percent. This reduction in error is present, albeit smaller, even at higher speeds. At the highest speed of 2640 μm/s, there was still around a 40% decrease in position estimation error during the occlusion trials. The absolute position error decrease was around 3

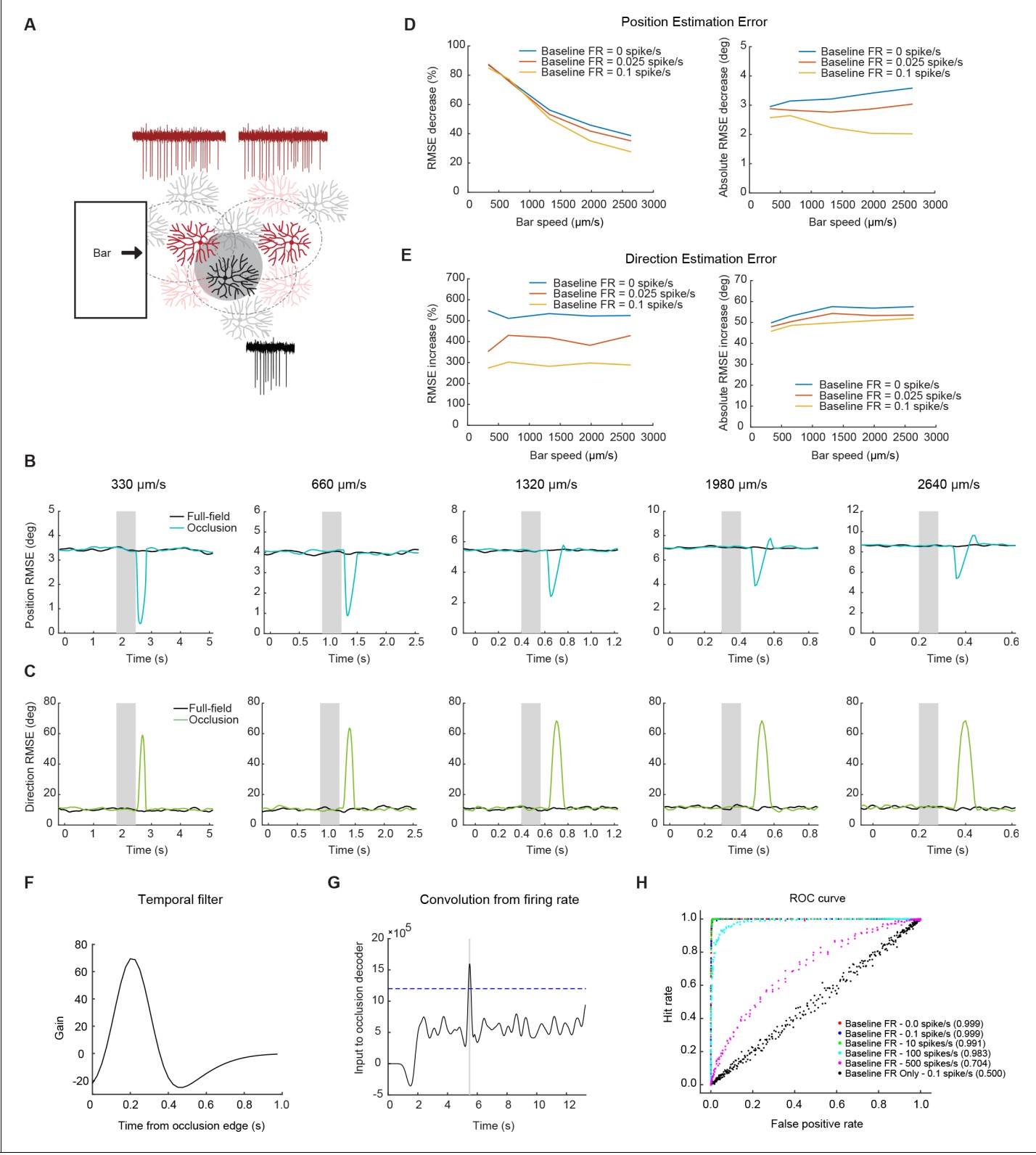

**Figure 7.** Null-direction spiking of On-Off DSGCs during the occluded motion stimulus improves position estimation. (**A**) Schematic of the DSGC population model along with example spike trains for two pDSGCs (red trace showing their preferred-direction responses) and an aDSGC (black trace showing its null-direction response). An occluder 220 μm wide (shaded circle) was placed near the middle of the population. (**B**) Example root-mean-square error in estimating the position of the light bar's leading edge for full-field (black) and occluded (blue) motion across different bar speeds.

*Figure 7 continued on next page*

*Figure 7 continued*

Shaded area indicates the time window during which the moving edge is behind the occluder. (C) Same as in B, but for direction estimation error. (D) Left: Position error percent decrease across speeds. Right: Absolute error in visual angle decrease across bar speeds. (E) Same as in D, but for direction estimation error. Right: Absolute error in motion direction angle. (F) Temporal filter for detecting occlusion response. (G) Convolution of firing rate from the temporal filter. Dashed line represents example threshold. (H) ROC curve with different levels of baseline firing for a bar of speed 330 μm/s. Legend shows performance level of decoder for each level of baseline firing.

The online version of this article includes the following source data and figure supplement(s) for figure 7:

**Figure supplement 1.** Model fits to experimental recordings.

**Figure supplement 1—source data 1.** Model fits to experimental recordings.

degrees of visual angle, or around half the receptive field size of an On-Off DSGC, across bar speeds in models with low levels of background noise. Therefore, at the site of motion interruption, the DSGC population response transiently prioritizes the encoding of the emerging moving object position over its motion direction.

Because the displaced excitatory receptive field also induced differential firing responses to stationary spots presented outside of the cell's dendritic field (*Figure 1*), we asked whether the displacement was useful for detecting non-directional contrast changes for a small stationary spot. Receptive field mapping experiments showed that the spiking receptive field is displaced toward the preferred side. However, the total diameter of the asymmetric receptive field is around 220 μm, which is larger than the average dendritic span of pDSGCs (*Figure 1—figure supplement 1H*). We analyzed whether a displaced receptive field would benefit the position estimation using non-directional contrast change signals. We found that the accuracy of estimating position from contrast changes is the same for populations with asymmetric receptive fields and populations with symmetric receptive fields. The detection of the contrast change would only improve if the receptive field size decreased (data not shown). Directed motion is required for an improvement in position estimation because a null-direction signal can only occur in a small region on the preferred side. Therefore, the synchrony of the null-direction and preferred-direction signals within the population yields a more spatially constrained signal necessary for fine spatial discrimination of moving stimuli.

The synchronous firing may, itself, also be a useful alarm signal that triggers processing downstream of the retina, independent of the precise position information it might additionally convey (*Ishikane et al., 2005*). During smooth motion, only one subtype of DS cell would respond to a bar moving across the visual scene. However, synchronous firing of two oppositely tuned DS cells would occur to represent interrupted or emergent motion. The synchronous firing of two DSGC subtypes can be a unique signature of encoding interrupted motion, which can differ from the encoding of non-motion contrast signals where the recruitment of all four DSGC subtypes would be expected. To investigate whether the synchronous firing during occlusion trials can be read out from a population response, we utilized a coincidence decoder to determine whether full-field motion trials can be distinguished from occluded motion trials. A filter was fit to the null-direction spiking response from the occlusion trials (*Figure 7F*), and when the convolution of the firing rate with the filter rose above a threshold level (*Figure 7G*), the detector would identify an occlusion trial.

Evaluating the decoder performance showed that the decoder is highly successful at identifying occlusion trials in conditions with low background firing rates. In conditions with background firing rates of 10 spikes/s or less, the decoder correctly identified occlusion trials with more than 99% accuracy (*Figure 7H*). This coincidence detection model suggests that the synchrony of preferred-direction responses from one DSGC subtype and the null-direction responses from the opposite DSGC subtype during interrupted motion conditions can be easily detected. This salient synchrony signal can potentially inform downstream visual areas of the type of motion that is occurring or provide an alarm signal to the animal that there is an unexpected change in their environment.

## Discussion

Our study reveals a new form of asymmetry in the direction-selective circuit: a spatial displacement of glutamatergic inputs to the preferred side of the On-Off pDSGCs due to a non-uniform distribution of synaptic conductances across the pDSGC dendritic span. The impact of this displaced excitation on DSGC spiking is demonstrated by using moving stimuli with interrupted trajectories, a

feature abundant in natural scenes. In contrast to full-field continuous motion which maximizes the contribution of SAC-mediated null-direction inhibition, occluded motion stimuli reduce the contribution of SAC-mediated inputs to allow the excitatory receptive field to dominate the spiking response in a non-directional manner when only part of the RF is stimulated. Therefore, an On-Off DSGC's response to occluded motion stimuli is determined both by how much of the receptive field is activated and where that activation occurs.

The non-isotropic excitation of the pDSGC alludes to a more sophisticated set of signaling mechanisms from the bipolar cell population to pDSGCs. A detailed explanation of the glutamatergic RF displacement awaits future studies. The gradient of glutamatergic current density across the pDSGC dendritic span may arise from a number of possible scenarios including varying strengths of individual glutamatergic synapses, heterogeneous membrane properties across pDSGC dendritic field, and contribution from Vglut3 +amacrine cells (*Franke et al., 2017*; *Kim et al., 2015*; *Lee et al., 2016*; ).

It is intriguing to speculate how the displaced excitatory receptive field properties of On-Off DSGCs can influence downstream computation. Previous theoretical analyses have addressed the encoding of motion direction by DSGCs at the population level (*Fiscella et al., 2015*; *Zylberberg et al., 2016*). In this study, we explored a hypothesis that On-Off DSGC population activity contains information about both the direction and the location of a moving object. When motion trajectories are not continuous, the null-direction responses from cells near the occlusion edge transiently improves the encoding of location at the expense of direction encoding. We speculate that the trade-off between positional and directional encoding, which occurs when an object emerges from behind another object, may reflect the animal's greater need for positional information of the emerging object than the direction in which it is moving. Additionally, the synchronous response of null-direction and preferred-direction spiking can potentially provide a salient alarm signal for discontinuous motion, which may help the animal quickly attend to the site of the change. Considering the population activity of multiple subtypes of On-Off DSGCs after an interruption in the motion allows for the encoding of more information than motion direction. Our findings that the population activity across cell types can help resolve ambiguities in single-cell responses share a common theme with previous modeling experiments from *Kühn and Gollisch, 2019*, which show that multiple DSGC subtypes with different motion direction tuning in the salamander retina are needed to isolate motion-related information from confounding contrast signals under complex texture motion. Our current investigation adds to the accumulating evidence that retinal population activity across multiple subtypes enhances decoding of visual features from ambiguous, multiplexed signals.

Elements of the visual processing scheme implicated in our study parallel those in other visual areas and species. For example, specific stimulus patterns, such as occluded motion in our study and the 'reverse-phi' illusory motion with alternating contrast polarities studied in flies, trigger null-direction responses in direction-selective neurons to serve context-specific encoding tasks that may be beneficial for extracting visual information from more complex natural scenes (*Salazar-Gatzimas et al., 2018*; *Agrochao et al., 2020*). As another example, both previous studies in flies and our study indicate that local motion detection can be carried out by neuronal populations with wider receptive fields (*Fisher et al., 2015*). In our model, partial RF activation on the preferred side of the pDSGC generates a non-directional, local contrast response that may contribute to a salient population signal from the retina to the brain to alert the animal about emerging motion. Interestingly, local contrast response properties of direction-selective neurons in the fly visual system have been shown to profoundly modulate their motion computations and contribute to visually guided behavior (*Clark et al., 2014*; *Drews et al., 2020*; *Matulis et al., 2020*).

In contrast to the mammalian On DSGC that encodes global motion during optic flow and participates in the optokinetic reflex, On-Off DSGCs are considered encoders of local motion, and project to the superficial layer of the superior colliculus (SC) and the shell region of the dorsal lateral geniculate nucleus (dLGN) (*Cruz-Martín et al., 2014*; *Huberman et al., 2009*; *Kay et al., 2011*; *Rivlin-Etzion et al., 2011*). In the SC, On-Off DSGC inputs give rise to the direction selectivity of postsynaptic collicular neurons (*Shi et al., 2017*), indicating that these collicular neurons do not receive retinal inputs from a broad range of RGC types, but specifically from On-Off DSGCs. Since the superficial layer of the SC is well recognized for its roles in encoding spatial locations and instructing stimulus-directed defensive and prey behaviors (*Basso et al., 2021*; *Ito and Feldheim, 2018*), the

encoding of the spatial location of an emerging moving object by On-Off DSGCs may benefit rapid sensorimotor decisions that involve collicular circuitry.

It is likely that other ganglion cell types also participate in fine spatial discrimination. For example, certain types of small receptive field RGCs such as W3 RGCs and HD-RGCs may also be well-suited to encode object location (*Jacoby and Schwartz, 2017*; *Kim et al., 2010*; *Zhang et al., 2012*). It is worth noting that W3 RGCs are activated only in specific instances where the background is completely uniform, whereas On-Off DSGCs can be activated in a wide range of visual environments, including environments with noisy backgrounds (*Chen et al., 2016*; *Chen et al., 2020*). HD-RGCs also have small receptive fields, and computational modeling experiments have shown that the errors in object location between On-Off DSGCs after motion interruption and HD-RGCs are very similar in scale (*Jacoby and Schwartz, 2017*). However, given the divergent and type-specific central projection patterns of mouse ganglion cell types (*Dhande et al., 2015*; *Ellis et al., 2016*; *Martersteck et al., 2017*), the position information encoded by other position encoders such as W3 and HD-RGCs may not be available to the specific downstream circuits that receive On-Off DSGC inputs. Our modeling study suggests that the population response of On-Off DSGCs after a motion interruption helps On-Off RGCs achieve the same performance as other ganglion cell populations implicated in fine spatial discrimination.

Given the many outstanding questions on the retinorecipient circuitry, On-Off DSGCs may participate in different visual processing tasks compared to other ganglion cell types by projecting to different areas for implementing different responses (*Kay et al., 2011*; *Sanes and Masland, 2015*). Or, they could provide complementary information about the spatial location of moving objects when considered with other RGC populations. In either scenario, our theoretical analysis indicates that the information of spatial location is contained within the On-Off DSGC population response, and that it is possible that higher visual centers stand to benefit from this information. Ultimately, elucidating the roles of diverse RGC types in motion encoding requires a thorough understanding of visual signal transformations along the processing pathways and how the visual system instructs visually guided behavior.

## Materials and methods

### Key resources table

| Reagent type (species) or resource | Designation | Source or reference | Identifiers | Additional information |
|---|---|---|---|---|
| Gene (*M. musculus*) | 129S6-*Chat*$^{tm2(cre)Lowl}$/J | The Jackson Laboratory | RRID:IMSR_JAX:006410 | |
| Gene (*M. musculus*) | 129S6-*Gt(ROSA)26 Sor*$^{tm9(CAG-tdTomato)Hze}$/J | The Jackson Laboratory | RRID:IMSR_JAX:007909 | |
| Gene (*M. musculus*) | *Slc17a6*$^{tm2(cre)Lowl}$/J | The Jackson Laboratory | RRID:IMSR_JAX:012898 | |
| Gene (*M. musculus*) | B6J.Cg-*Gt(ROSA)26 Sor*$^{tm95.1(CAG-GCaMP6f)Hze}$/MwarJ | The Jackson Laboratory | RRID:IMSR_JAX:028865 | |
| Chemical compound, drug | Dihydro-b-erythroidine hydrobromide | Tocris | Cat#2349 | |
| Chemical compound, drug | SR 9551 hydrobromide | Tocris | Cat#1262 | |

### Animals

129S6-*Chat*$^{tm2(cre)Lowl}$/J mice (Stock No: 006410) and 129S6-*Gt(ROSA)26Sor*$^{tm9(CAG-tdTomato)Hze}$/J mice (Stock No: 007909) were acquired from the Jackson Laboratory. *Drd4*$^{GFP}$ mice were originally developed by MMRRC (http://www.mmrrc.org/strains/231/0231.html) and were backcrossed to the C57BL/6 background. These lines were crossed so that both pDSGCs and SACs were labeled. *Slc17a6*$^{tm2(cre)Lowl}$/J mice (Stock No: 016963) and B6J.Cg-*Gt(ROSA)26Sor*$^{tm95.1(CAG-GCaMP6f)Hze}$/MwarJ mice (Stock No: 028865) were acquired from Jackson Laboratory and were crossed and used

for calcium imaging experiments. Mice of ages P21 – P60 of either sex were used. All procedures regarding the use of mice were in accordance with the University of Chicago Institutional Animal Care and Use Committee (ACUP# 72247), Institutional Biosafety Committee (IBC# 1214), and with the NIH Guide for the Care and Use of Laboratory Animals and the Public Health Service Policy.

## Whole-mount retina preparation

Mice were dark adapted for 1 hr and then were anesthetized with isoflurane and euthanized by decapitation. Retinas were isolated at room temperature in oxygenated Ames' medium (Sigma-Aldrich, St. Louis, MO) under infrared illumination. The retinas were separated into dorsal and ventral halves and were mounted ganglion-cell-layer up on top of a ~ 1.5 mm$^2$ hole in a small piece of filter paper (Millipore, Billerica, MA). The orientation of the posterior, anterior, inferior, and superior directions were noted for each piece. During the experimental day, the mounted retinas were kept in darkness at room temperature in Ames' medium bubbled with 95% $O_2$/5% $CO_2$ until use (0–8 hr).

## Visual stimulation

A white organic light-emitting display (OLEDXL, eMagin, Bellevue, WA; 800 × 600 pixel resolution, 60 Hz refresh rate) was controlled by an Intel Core Duo computer with a Windows seven operating system and was presented to the retina at a resolution of 1.1 µm/pixel. Visual stimuli were generated using MATLAB and the Psychophysics Toolbox, and were projected through the condenser lens of the two-photon microscope focused on the photoreceptor layer.

For the peripheral spot experiments, stationary spots of size 110 µm diameter were presented 165 µm away from the soma of the recorded pDGSC for 2 s. At this distance, these spots were presented outside of the dendritic span of the majority of recorded pDSGCs. These spots were presented in eight different locations pseudorandomized around the recorded pDSGC, with three repetitions each.

For moving bar experiments, a bright moving bar with dimensions 110 µm (width) X 880 µm (length) and speed 330 µm/s moved in eight different pseudorandomized directions across the receptive field of the pDSGC, with three to six repetitions each. The occluded bar stimulus contains a moving bar of the same dimensions which moves into and out of a central occluder with a diameter size of 220 µm. The size of the occluder was large enough to obscure the entire dendritic span for the majority of recorded pDSGCs. Occluded bar stimuli of different contrasts were used. The population vector model in *Figure 7* required the use of bar and occluded bar stimuli at speeds of 330 µm/s, 660 µm/s, 1320 µm/s, 1980 µm/s, and 2640 µm/s.

For the emergent moving bar experiments where the bar emerged from different locations of the receptive field, a bright moving bar with dimensions 110 µm (width) X 440 µm (length) and speed 330 µm/s could start moving pseudorandomly from 13 different locations across the receptive field (*Figure 5*). The bar could move in the preferred or null direction of the recorded cell.

For the receptive field mapping experiments measuring spiking activity, bright spots with a 60 µm diameter were presented across the receptive field of the recorded pDSGC. They were shown for three to four repetitions with a duration of 400 ms in a random sequence within an 11 × 11 grid, covering a total area of 660 × 660 µm$^2$. For the receptive field mapping experiments measuring EPSC activity, bright spots with a 20 µm diameter were presented across the pDSGC and were shown for three to four repetitions with a duration of 400 ms in a random sequence within an 11 × 11 grid, covering a total area of 220 × 220 µm$^2$. To examine the kinetics of EPSCs, bright spots with either a 20 µm diameter (for DHβE and gabazine experiments) or a 60 µm diameter (for DHβE experiments) were shown across the receptive field in a random sequence within an 11 × 11 grid for three to four repetitions. Spots were shown with a duration of 400 ms.

The light intensity for bright peripheral spot experiments, the moving bar and occlusion experiments, and the receptive field mapping experiments was ~1.6×10$^4$ R*/rod/s.

## Targeting cells for electrophysiology

Cells were visualized with infrared light (>900 nm) and an IR-sensitive video camera (Watec). DSGCs were targeted with the aid of two-photon microscopy in *Drd4-GFP* mice. Cell identity was confirmed physiologically by extracellular recordings of responses to moving bars and/or by filling the cell with 25 µM Alexa 594 (Life Technologies) to show bistratified dendritic morphology. Because the tissue

was aligned on the filter paper, we confirmed that the preferred direction of the spiking responses to moving bar aligned well with the anatomical posterior (visual field coordinate) or nasal (retinal coordinate) direction.

## Quantification of dendritic morphology

The On and Off layers of each cell was isolated from a z-stacked image in ImageJ. To calculate the radius, a contour of the dendritic span was made by drawing a boundary around the dendritic tips, and the area within the contour and resulting equivalent radius was calculated. For calculating dendritic length, the dendrites from the z-stack projection was traced in NeuronStudio. The two-dimensional trace was then exported to MATLAB. Dendritic length was calculated in MATLAB using a custom-written code. We subdivided the cell into eight sectors for analysis. Sectors extended out from the soma and were aligned to the spot stimulus such that each spot was at the center between each sector's radial boundaries. When computing normalized vector sums (*Figure 2B–C*, *Figure 2— figure supplement 1A-B*), total dendritic length in each sector was normalized by the total dendritic length of the cell.

## Serial electron-microscopic analysis

We reconstructed DSGCs and starburst amacrine cells from a previously published SBEM dataset (*Ding et al., 2016*). These cell types were easily recognizable from their characteristic dendritic arbors and patterns of stratification. We inferred the preferred direction of motion of the DSGCs, relative to the boundaries of the volume from the orientation of SAC dendritic inputs onto the DSGC dendrites, as in *Briggman et al., 2011*. SAC dendrites preferentially form GABAergic synapses with DSGCs if their orientation, from soma to dendritic tip, corresponds to the null direction of the DSGC. Neither the eye of origin nor the retinal location of the sample was recorded when this sample was acquired, but it was possible to infer the ventral direction within the volume from reconstructions of multiple members of several other RGC types with strong dendritic asymmetries along the dorsoventral axis, including Jam-B RGCs (*Kay et al., 2011*) and F-RGCs (*Rousso et al., 2016*). One of the four On-Off DSGC types had an inferred preference for the ventral direction. Thus, we infer that the two types with preferred directions 90 degrees away from this ventral-motion-preferring DSGC type must therefore have been tuned to the horizontal motion axis. We could think of no way to distinguish which of these types prefers anterior motion and which prefers posterior motion.

Ribbon synapses from On bipolar cells onto the dendrites of these three DSGCs were mapped manually. The branching patterns of all cells were reconstructed using skeletons and synapses were marked with single nodes using the Knossos software package (https://knossostool.org/).

To measure local ribbon density across the DSGC dendritic field, the dendritic span was divided into individual squares with $10 \times 10$ μm (*Figure 4—figure supplement 1*) or $20 \times 20$ μm (*Figure 4*) dimensions. Total dendritic length in each square was calculated in Matlab with a custom-written code. The total number of ribbon synapses in each square was also quantified. Ribbon density was calculated by dividing the ribbon synapse number by the total dendritic length in the appropriate squares.

## Electrophysiology recordings

Recording electrodes of 3–5 MΩ were filled with a cesium-based internal solution containing 110 mM CsMeSO4, 2.8 mM NaCl, 4 mM EGTA, 5 mM TEA-Cl, 4 mM adenosine 5′-triphosphate (magnesium salt), 0.3 mM guanosine 5′-triphosphate (trisodium salt), 20 mM HEPES, 10 mM phosphocreatine (disodium salt), 5 mM N-Ethyllidocaine chloride (QX314),filled with a cesium-based internal solution containing 110 mM CsMeSO4, 2.8 mM NaCl, 4 mM EGTA, 5 mM TEA-Cl, 4 mM adenosine 5′-triphosphate (magnesium salt), 0.3 mM guanosine 5′-triphosphate (trisodium salt), 20 mM HEPES, 10 mM phosphocreatine (disodium salt), 5 mM N-Ethyllidocaine chloride (QX314), and 0.025 mM Alexa 594, pH 7.25. Retinas were kept in oxygenated Ames' medium with a bath temperature of 32–34°C.

Data were acquired using PCLAMP 10 and a Multiclamp 700B amplifier (Molecular Devices, Sunnyvale, CA), low-pass filtered at 4 kHz and digitized at 10 kHz. Light-evoked EPSCs were isolated by holding cells at −60 mV after correction for the liquid junction potential (~10 mV).

To isolate the contribution of synaptic inputs, a host of pharmacological agents were perfused in the bath during electrophysiology recordings. 8 µM Dihydro-b-erythroidine hydrobromide (DHβE; Tocris, Cat#2349); 10 µM SR 9551 hydrobromide (gabazine; Tocris, Cat #1262); 100 µM Meclofenamic acid sodium salt (MFA; Sigma-Aldrich. Cat#M4531); 1 µM Strychnine (Sigma-Aldrich, Cat#S0532); 50 µM TPMPA (Tocris, Cat#1040); 50 µM Carbenoxolone disodium (Tocris, Cat#3096).

## Data analysis of electrophysiological recordings

Spiking data from loose-patch recordings were analyzed using custom protocols in MATLAB. The number of spikes evoked by the response to the peripheral spots were quantified in MATLAB and averaged across three repetitions in eight spatial locations. Light-evoked EPSC responses to peripheral spots were obtained as well, and three repetitions of EPSC traces were averaged to obtain the mean peak amplitude and charge transfer response to each condition. The RSI is determined by (Max - Opp)/(Max + Opp) where Max is the number of spikes, peak amplitude or charge transfer in the region of maximal activation using the peripheral spot stimulus, and Opp is the response in the region directly opposite to the Max region. EPSC parameters of latency, rise time (10–90%), and decay time (90–10%) were quantified to compare the whole cell kinetics after administration of DHβE and gabazine to the control condition (*Figure 1—figure supplement 1E*).

For the receptive field mapping experiments, the light-evoked EPSCs in response to spots sized either 20 µm or 60 µm in diameter were obtained and averaged across three to four repetitions. The displacement of the spiking RF was determined by using 60 µm spots. The distance from the soma to the edges of the receptive field on the preferred side versus the null side were determined (*Figure 1—figure supplement 1H*).

For the receptive field mapping EPSC experiments using 20 µm spots, the center of mass of the receptive field was determined using the EPSC charge transfer or amplitude at each square in the 11 × 11 grid. The dendritic length was calculated at each point on the RF stimulus grid by summing the dendritic length in a circle 60 µm in diameter centered on the square of interest. For maximum vs. opposite region analysis and the preferred side vs null side analysis (*Figure 3E–F*), we considered the EPSCs in two 5 × 5 subsets of the total grid located opposite of each other. The preferred-null side of each cell was determined by spiking responses to the full-field moving bar, and the maximum-opposite axis was determined by EPSC charge transfer responses to the peripheral spot experiment performed in DHβE and gabazine. The distance from soma at each grid location was defined as the distance between the center of the square and the soma center.

To estimate the spatiotemporal profile across the maximum-opposite or the preferred-null axis for cells in DHβE and gabazine, the light-evoked EPSCs in response to small spots sized 20 µm in diameter were obtained and averaged across three to four repetitions (*Figure 4F–H*). To estimate the spatiotemporal profile across the preferred-null axis for cells in DHβE only, the light-evoked EPSCs in response to small spots sized 60 µm in diameter were obtained and averaged across three repetitions (*Figure 4—figure supplement 1C-D*). For both experiments, EPSC parameters of latency, rise time (10–90%), and decay time (90–30%) were quantified. The preferred-null axis of each cell was determined by spiking responses to the full-field moving bar, and the maximum-opposite axis was determined by EPSC charge transfer responses to the peripheral spot experiment performed in DHβE and gabazine. The parameters of latency, rise time, and decay time were averaged across the squares at equal distances along the cell's preferred-null axis or maximum-opposite axis.

Spiking data evoked by the moving bar and occlusion stimuli were quantified in MATLAB using three to six repetitions in eight different directions. Spiking data evoked by the bar starting in different positions were quantified in MATLAB across three repetitions in the preferred and null directions.

## Calcium imaging in posterior- and anterior-preferring On-Off DSGCs

Genetically encoded calcium indicator GCaMP6f was expressed in all RGCs by crossing *Slc17a6<sup>tm2 (cre)Lowl</sup>* mice and B6J.Cg-*Gt(ROSA)26Sor<sup>tm95.1(CAG-GCaMP6f)Hze</sup>*/MwarJ mice. GCaMP6f fluorescence from isolated retinas was imaged in a customized two-photon laser scanning fluorescence microscope (Bruker Nano Surfaces Division). GCaMP6 was excited by a Ti:sapphire laser (Coherent, Chameleon Ultra II, Santa Clara, CA) tuned to 920 nm, and the laser power was adjusted to avoid saturation of the fluorescent signal. Onset of laser scanning induces a transient response in RGCs

that adapts to the baseline in ~3 s. Therefore, to ensure the complete adaptation of this laser-induced response and a stable baseline, visual stimuli were given after 10 s of continuous laser scanning. To separate the visual stimulus from GCaMP6 fluorescence, a band-pass filter (Semrock, Rochester, MA) was placed on the OLED to pass blue light peaked at 470 nm, while two notched filters (Bruker Nano Surfaces Division) were placed before the photomultiplier tubes to block light of the same wavelength. The objective was a water immersion objective (60x, Olympus LUMPlanFl/IR). Time series of fluorescence were collected at 15–30 Hz.

We performed an initial direction selectivity test to identify posterior- and anterior-preferring On-Off DSGCs. We recorded GCaMP6f fluorescence from RGC somas within a 75 µm X 75 µm field of view while presenting a full-field moving bar visual stimulus (a bright moving bar 110 µm (width) X 880 µm (length) moving at 330 µm/s across a 660 µm circular mask diameter along eight different directions). At the onset of each moving bar sweep, a TTL pulse was triggered by the visual stimulus computer and recorded by the imaging software to correlate GCaMP6f signals with the direction of each moving bar. Immediately following acquisition of each time series stack, custom-written MATLAB scripts were used to extract fluorescence over time data from time-series images and sort calcium transient by direction of the moving bar. For each RGC soma, raw GCaMP6f fluorescence traces and tuning curves were plotted. On-Off DSGCs were identified by their characteristic singular-lobe directional tuning curves, DSI values $\geq$ 0.3, and two fluorescence peaks time-locked to the leading (On) and trailing edge (Off) of the moving bar. On-Off DSGCs with preferred directions along the posterior-anterior axis were then selected for further imaging.

Once an On-Off DSGC of interest was identified, the visual stimulus was centered to the soma of that cell and a new field of view was drawn to enclose this cell and some background with no GCaMP6f fluorescence. Full-field and occlusion moving bar visual stimulus were presented to the cells as described above (eight directions, three to four repetitions). Time series data was collected and subjected to offline analysis.

## Imaging analysis for calcium imaging

Analysis was performed using ImageJ and MATLAB. Regions of interest (ROIs) corresponding to DSGC soma and background were manually selected in ImageJ. The fluorescent time course of each ROI was determined by averaging all pixels within the ROI for each frame. The fluorescence of the background region was subtracted from the raw fluorescent signal of the soma ROIs at each time frame. The visual stimulus included a 3–4 s intersweep interval between the end of one sweep and the start of another. Fluorescence intensities during these intersweep intervals were used to create a baseline (F0) trace for each ROI by fitting either a single- or two-term exponential decay function. Fluorescence measurements were then converted to ΔF/F0 values by calculating ΔF=(F−F0)/F0 for every datapoint. The transformed traces were then smoothed using an average sliding window of 4 datapoints. ΔF/F0 traces were clipped, sorted by visual stimulus direction (0, 45, 90, 135, 180, 225, 270, and 315 degrees), and averaged over three to four trials. Prior to further analysis, ROIs were subjected to a response quality test QI = Var[Avg. Resp]/Avg(Var[R(t)])$\geq$0.45 to ensure consistency across trials. Responses to the full-field and occlusion moving bars were broken up into On and Off components according to the circular mask entrance and exit times of the leading and trailing edge, respectively. Peak, area ΔF/F0, and time of peak values were calculated for On, Off, and the full trace along all eight directions. Direction selectivity index (DSI), vector sum, and preferred direction were calculated for both On and Off components.

## Statistical analysis

Grouped data are presented as mean ± SEM. The Kolmogorov-Smirnov test was used to test data for normality. Student's t-test was used for statistical comparisons of paired samples in *Figures 1* and *2*. One-way analysis of variance was performed on grouped data in *Figures 5* and *6* and subjected to Bonferroni correction.

For the EPSC/dendritic length vs distance from soma experiments, we performed linear regression analysis using an additional categorical predictor variable indicating the maximum-opposite or the preferred-null side. The p-value associated with interaction term (distance*region) in the resulting model was used to determine whether the slopes are significantly different between the two regions. For the EPSC vs dendritic length experiments, we again performed linear regression with an

additional categorical predictor variable indicating the maximum-opposite or preferred-null region. The p-value associated with the categorical predictor in the resulting model was used to determine whether the y-intercepts were significantly different between the two regions. The p-value associated with the categorical predictor was used to determine whether the y-intercepts were significantly different between regions. The number of branches in each square of the grid was determined by a custom MATLAB code.

For the kinetic analyses of EPSC parameters in the receptive field mapping experiments of latency, rise time, and decay time, we performed linear regression analysis to determine whether a statistically significant linear relationship exists between the distance from the soma and each EPSC parameter.

For all data sets, p<0.05 was considered significant. *p<0.05; **p<0.01; ***p<0.001.

## Experimental parameters for population model

Experimental data for the spiking response to full-field (73 cells) and occlusion (69 cells) stimuli moving with a constant speed of 330 µm/second were obtained. Three to six repetitions were obtained for each full-field or occlusion protocol. The baseline firing rate for each repetition was obtained by binning the spiking response in 25 ms time bins and taking the maximum firing rate during a silent period where no stimulus was displayed, and the baseline firing rate was averaged across all repetitions.

To model the spiking response of the pDSGCs, we binned the spikes evoked by the On response to the motion stimulus in 25 ms time bins and plotted the PSTHs for all eight motion directions. Then, we fit a rectified sine wave to the PSTH of each pDSGC. We defined the threshold for above-baseline firing to be 4 SD above the baseline firing rate. The onset of the spiking response was determined by the time bin at which the firing rate exceeded the threshold and was immediately followed by a second above-threshold bin (Rate change method; *Levakova et al., 2015*). We inspected the spiking response onset times returned by our detection algorithm and manually adjusted the spiking response onset times for 3 out of 91 pDSGCs to match the experimental data.

Onset times of spiking responses to the full-field bar moving in the preferred direction ±45 degrees were similar to those of the preferred-direction response (*Figure 7—figure supplement 1A*, left). Likewise, onset times of spiking responses to the occlusion stimulus moving in the null direction ±45 degrees were similar to those of the null-direction response (*Figure 7—figure supplement 1A*, right). Therefore, we included onset times of spiking responses to motions in the directions ± 45 degrees from the preferred-null motion axis in our analysis. Four-parameter beta distributions were fit to histograms of spiking response onset times for preferred- and null-direction motions (*Figure 7—figure supplement 1F*). The four parameters included two shape parameters and two parameters that specify the minimum and maximum of the distribution's range.

To determine the offset of the spiking response, we used the algorithm for identifying spiking response onset. Unlike the protocol for determining spiking response onset, however, the detection algorithm started from the most recent time bin and traversed backwards in time. The offset of the spiking response was defined to be the time bin at which the firing rate exceeded the threshold in two out of three consecutive time bins. For each pDSGC, we calculated the spiking response duration by finding the difference between the spiking response onset and offset times. Furthermore, we calculated the linear correlation between spiking response duration and onset (*Figure 7—figure supplement 1B*). *Figure 7—figure supplement 1B* shows that using only spiking responses to motion along the preferred-null motion axis and using spiking responses to motion along the preferred-null motion axis as well as motion in the directions 45 degrees away from the preferred-null motion axis yielded consistent results. We also calculated the linear correlation between peak firing rate and spiking response onset, but the correlation was not significant for the full-field protocol (*Figure 7—figure supplement 1C*).

## Motion direction tuning curves

We computed motion direction tuning curves for all pDSGCs exposed to the full-field moving bar stimulus (73 cells) using the CircStat toolbox in MATLAB developed by *Berens, 2009*. The height of each tuning curve was given by the total spike count evoked during the presentation of the full-field moving bar stimulus. We fit a Gaussian function to the histogram of the tuning curve widths

(*Figure 7—figure supplement 1G*). *Figure 7—figure supplement 1H* shows all the normalized motion direction tuning curves. The heights of the tuning curves were rescaled by dividing by their peaks and the widths of the tuning curves were rescaled by dividing by their angular deviation, which is the square-root of twice the circular variance. The normalized tuning curves were fit to a one-term Gaussian model (*Figure 7—figure supplement 1H*).

## Speed tuning analysis

To investigate whether the null-direction response remains robust at higher stimulus speeds, experimental data for the spiking response to full-field and occlusion stimuli moving at 660, 1320, 1980, and 2640 µm/s were further collected. We analyzed how the spike count, onset, and duration of the null-direction response changed across bar speeds (*Figure 7—figure supplement 1I-K*). Spiking response onset and offset were calculated using the same method as before. Linear fits were performed on spiking response onset data across speeds for both full-field and occlusion protocols (*Figure 7—figure supplement 1J*). Spiking response durations were first normalized by the response duration when the bar speed was at 330 µm/s and then fit to power-law functions (*Figure 7—figure supplement 1K*). Simulation parameters for stimulus speeds higher than 330 µm/s were adjusted according to the fit functions in *Figure 7—figure supplement 1J and K*.

## Two-dimensional population model

In our computational model, we arranged two populations (left motion-preferring and right motion-preferring) of DSGCs in a two-dimensional array. Each population had 1,000 cells. Within each population, horizontal and vertical distances between nearest neighbors were Gaussian distributed, with a mean of 39 µm and a SD of 16 µm (*Huberman et al., 2009*). The spatial positions of the DSGCs between the two populations were uncorrelated.

Each DSGC's mean spiking response to the moving edge was modeled as a rectified sine wave. The amplitude of the sine wave was given by the peak firing rate, while the period and phase were determined by the spiking response duration and onset time, respectively. For each spatial arrangement of DSGCs, we sampled peak firing rates directly from our experimental data. We sampled spiking response onset times from our four-parameter beta distributions (*Figure 7—figure supplement 1F*) and determined the spiking response durations by finding the linear correlation between the two (*Figure 7—figure supplement 1B*). For simulations with bar stimuli moving at speeds higher than 330 µm/s, spiking response onset times and durations were modified according to the speed of the bar (*Figure 7—figure supplement 1J and K*).

Noise was introduced into the preferred directions of the DSGCs so that they were not all perfectly aligned with the left/right motion axis. To determine the degree of jitter in a DSGC's preferred direction, we sampled from a uniform distribution ranging from −14.1 degrees to +14.1 degrees, where 0 degrees represented a preference for motion directly along the left/right motion axis (*Fiscella et al., 2015*).

To determine the motion direction tuning width of each simulated DSGC, we sampled circular variances from a Gaussian distribution fit to the histogram of circular variance of the tuning curves from our experimental data (*Figure 7—figure supplement 1G*). We scaled the collapsed tuning curve (*Figure 7—figure supplement 1H*) by the sampled tuning width and peak firing rate to obtain the tuning curve. We used the motion direction tuning curve to adjust the peak firing rate of the spiking response according to the jitter in the preferred direction alignment.

We simulated the DSGC population response to a moving edge traveling from left to right at a constant speed. To simulate the occlusion protocol, we introduced an occlusion 220 µm in diameter whose position in space was fixed at 1800 µm along the horizontal axis and 800 µm along the vertical axis (approximately in the center of the two-dimensional array). For each spatial arrangement of DSGCs, the simulation was repeated 10 times. The spiking response of each DSGC was discretized in time.

At each 10 ms time bin, the firing rate given by the rectified sine wave fit was converted to a mean spike count. The number of spikes generated by a DSGC was obtained by sampling from a Gaussian distribution with this mean and a sub-Poisson, constant variance of 0.4. The sub-Poisson noise was determined from our experimental data by analyzing the variance of the spiking responses to 6 repetitions of the full-field moving bar (10 cells) and the occlusion stimulus (9 cells) (*Figure 7—*

*figure supplement 1D*). The spatial positions of the DSGCs were shuffled in each simulation block for a total of 100 blocks with 10 repetitions in each block.

## Position and direction decoding

The spatial position and motion direction of the moving bar's leading edge were estimated via a labeled-line decoder (*Dayan and Abbott, 2001*). At each time point, the position estimate or direction $x$ was given by the weighted average of the DSGCs' RF center positions or preferred directions

$$\hat{x} = \frac{\sum_i r_i \tilde{x}_i / w_i^2}{\sum_i r_i / w_i^2}$$

where $r_i$ is the firing rate of the $i^{th}$ cell and $\tilde{x}_i \sim \mathcal{N}(x_i, w_i^2)$ where $x_i$ is the RF center position or the preferred direction and $w_i$ is the RF width (radius) or the motion direction tuning width of the $i^{th}$ cell. For position decoding, the RF width was taken to be the 1 SD boundary of the Gaussian center profile (*Chichilnisky and Kalmar, 2002*). RF widths were obtained by scaling the dendritic field radii by 1.25. Dendritic field radii were obtained by sampling from Gaussian distribution with $\mu = 88$ μm and $\sigma = 14.8$ μm. Position labels of the cells were determined by considering the extent of spatial displacement on the preferred side. Errors are reported as the root mean-square-error in the position or direction estimate (*Figure 7F-E*).

## Coincidence detection

To assess the salience of the synchronous firing between two oppositely tuned subtypes of DSGCs, we constructed a coincidence decoder that consisted of a convolution with a temporal filter and a threshold operation, similar to *Schwartz et al., 2007*. At each time point, a difference-of-Gaussian temporal filter (*Figure 7F*) was convolved with the simulated DSGC population firing-rate activity. When the output of the convolution exceeded the threshold, the decoder determined that a coincidence of spiking activity between DSGC subtypes has occurred. For correct detections, the output must exceed the threshold during a time window of 125 ms around the occlusion event when the bar emerges from behind the occluder (*Figure 7G*). All above-threshold outputs outside of the time window were marked as false alarms. By varying the threshold across multiple simulations, we computed the receiver operator characteristic (ROC) (*Green and Swets, 1966*). We quantified the performance of the decoder under different levels of background firing noise using the area under the ROC curve (*Figure 7H*).

## Acknowledgements

We thank Chen Zhang for managing the mouse colony and Dr. John Maunsell, Dr. Siwei Wang, Mathew Summers, Malak El-Quessny, and Benjamin Hoshal for advice on the manuscript. This work was supported by NIH R01 NS109990 and the McKnight Scholarship Award to WW, NSF GRFP DGE-1746045 to JD, NIH F31 EY029156 to HEA, NSF Career Award 1652617 and the Physics of Biological Function PHY-1734030 to SEP, and NIH RO1 EY012793 to DB.

## Additional information

### Competing interests

Stephanie E Palmer: Reviewing editor, *eLife*. The other authors declare that no competing interests exist.

### Funding

| Funder | Grant reference number | Author |
| --- | --- | --- |
| NIH | R01 NS109990 | Wei Wei |
| McKnight Endowment Fund for Neuroscience | McKnight Scholarship Award | Wei Wei |
| NSF | GRFP DGE-1746045 | Jennifer Ding |

| NIH | F31 EY029156 | Hector Acaron Ledesma |
|---|---|---|
| NSF | Career Award 1652617 | Stephanie E Palmer |
| National Science Foundation | PHY-1734030 | Stephanie E Palmer |
| NIH | RO1 EY012793 | David M Berson |

The funders had no role in study design, data collection and interpretation, or the decision to submit the work for publication.

### Author contributions

Jennifer Ding, Conceptualization, Formal analysis, Validation, Investigation, Visualization, Methodology, Writing - original draft, Writing - review and editing; Albert Chen, Conceptualization, Software, Formal analysis, Investigation, Visualization, Methodology, Writing - original draft, Writing - review and editing; Janet Chung, Hector Acaron Ledesma, Mofei Wu, Investigation, Methodology; David M Berson, Methodology; Stephanie E Palmer, Conceptualization, Software, Supervision, Funding acquisition, Visualization, Writing - original draft, Writing - review and editing; Wei Wei, Conceptualization, Resources, Supervision, Funding acquisition, Investigation, Methodology, Writing - original draft, Project administration, Writing - review and editing

### Author ORCIDs

Jennifer Ding (iD) https://orcid.org/0000-0003-2282-6615
Albert Chen (iD) http://orcid.org/0000-0002-9306-8703
Stephanie E Palmer (iD) http://orcid.org/0000-0001-6211-6293
Wei Wei (iD) https://orcid.org/0000-0002-7771-5974

### Ethics

Animal experimentation: All procedures regarding the use of mice were in accordance with the University of Chicago Institutional Animal Care and Use Committee (IACUC) (ACUP protocol 72247) and with the NIH Guide for the Care and Use of Laboratory Animals and the Public Health Service Policy.

### Decision letter and Author response

Decision letter https://doi.org/10.7554/eLife.68181.sa1
Author response https://doi.org/10.7554/eLife.68181.sa2

## Additional files

### Supplementary files

- Transparent reporting form

### Data availability

Data available on Dryad Digital Repository (http://doi.org/10.5061/dryad.vq83bk3s8). Source data files have been provided for all main text and supplementary figures.Code for model available on Github: https://github.com/jnnfr-ding/Occlusion-model, copy archived at swh:1:rev: 123261cdb72251e03cac9654713d17c4537d23a7.

The following dataset was generated:

| Author(s) | Year | Dataset title | Dataset URL | Database and Identifier |
|---|---|---|---|---|
| Ding J, Chen A, Chung J, Wu HM, Berson DM, Palmer SE, Wei W | 2021 | Spatially displaced excitation contributes to the encoding of interrupted motion by the retinal direction-selective circuit | http://dx.doi.org/10.5061/dryad.vq83bk3s8 | Dryad Digital Repository, 10.5061/dryad.vq83bk3s8 |

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
