## [Decision Letter]

**Acceptance summary:**

This paper studies a non-directional signal generated in On-Off directionally selective ganglion cells. Through a combination of experiment and modeling, the paper supports a picture in which this non-directional signal helps signal the location of a moving object, particular as an object emerges from behind an occluder. This provides a nice example of how selectivity for multiple stimulus features can support an interesting circuit function.

**Decision letter after peer review:**

[Editors’ note: the authors submitted for reconsideration following the decision after peer review. What follows is the decision letter after the first round of review.]

Thank you for submitting your work entitled "Spatially displaced excitation contributes to the encoding of interrupted motion by retinal direction-selective circuit" for consideration by *eLife*. Your article has been reviewed by 3 peer reviewers, including Fred Rieke as the Reviewing Editor and Reviewer #1, and the evaluation has been overseen by a Senior Editor.

Our decision has been reached after consultation between the reviewers. Based on these discussions and the individual reviews below, we regret to inform you that your work will not be considered further for publication in *eLife*, at least in present form. We would be willing to consider a revised manuscript if you are able to strengthen the work along the lines detailed below; we recognize though that this would require considerable effort.

The reviewers were in broad agreement that the findings were interesting and that the experiments were well executed and clear. The main concern is that the paper does not provide either a definitive mechanistic insight into why excitatory input is asymmetric, or a definitive functional argument about the importance of this asymmetry. This concern is detailed in the individual reviews, and was a focus of the consultation among the reviews. To be considered further, the paper would need to be strengthened considerably in one of these directions.

*Reviewer #1:*

This paper describes a new finding about stimulus encoding in On-Off directionally selective ganglion cells. It is well established that these cells have spatially displaced inhibitory input from starburst amacrine cells, and that the spatial offset of inhibitory input contributes to the cells' selectivity for direction of motion. The work in this paper shows that the cells also have spatially offset excitatory input, and that this input can give rise to a non-directional response. Several functional roles are suggested for the non-directional response. I felt that the evidence for the non-directional response was strong, but that the connection to visual function was too preliminary.

Functional importance

The paper emphasizes the possible functional importance of the non-directional motion signal; this is a focus of the discussion, and is highlighted in both the abstract and introduction. I found this part of the paper less complete and convincing than the experimentally-driven results. Several issues contribute to this. One is that the contribution to identifying the position of a moving object is fairly modest. Another is that the impact of the non-directional component on other stimulus properties – e.g. the accuracy with which motion direction is encoded – is not explored. A third is that the position of a moving object is almost certainly encoded by multiple ganglion cell types, and hence the modest improvement in position encoding in the DS cell population may make even less contribution when the entire ganglion cell population is considered. A complete investigation of coding in the ganglion cell population is clearly too much, but a more balanced and complete consideration of the benefits and drawbacks of the mechanism described would strengthen the paper considerably.

*Reviewer #2:*

In this research, Ding and colleagues present evidence that the excitatory input to OO DS RGCs from bipolar cells is strongly asymmetric, with strong inputs occurring on the side opposite from the SAC inhibition. They performed careful studies to show that this was not due to spatial asymmetry in the DSGC morphology nor to ribbon synapse density. Using 'interrupted motion' stimuli, which are effectively local directional stimuli, they show that this asymmetry leads to a non-directional response on one side of the cell's RF. Last, they create a model to show that such firing patterns could be used to improve localization of edge position under the specific conditions of an edge emerging from behind an occlusion.

The work showing the asymmetry appeared careful, thorough, and well-done. The second half of the paper dealing with the functional consequences of this asymmetry left me with a few questions:

1) Throughout the paper, several experiments showed no changes when a mix of receptor antagonists was added to exclude SAC inhibition as the origin of these effects. But I did not find a positive control, showing that these antagonists had the desired effect. Later, in Figures 5CD, the remaining effect after application of these antagonists was cited as evidence that the excitational asymmetry was responsible for the effect; that interpretation is only valid if the drugs truly kill all SAC input to the DSGC. What if the drugs were not 100% effective? Relatedly, in the experiments in 5CD, the measured responses all *decrease* with the antagonists, an effect that seems surprising and is not explained. Connecting the asymmetry in excitation to the interrupted motion is central to this paper, so it should have strong support.

2) The measured functional results appear quite similar to results in Kuhn and Gollisch 2019, which is not cited in that context. That paper found that DSGCs responded to local contrast, not just motion, much like the results here, and suggested that oppositely tuned cells could be subtracted to eliminate this contaminating contrast signal or added to isolate the contrast signal. Here, the authors suggest a very similar use for these signals, albeit with a decoder of position and a focus on motion rather than contrast changes. (See line 528, where the authors suggest that this position-direction hypothesis is new. See also line 537: or could not be salient, if there's any kind of downstream opponent subtraction, as in primate MT.)

3) The interrupted motion stimuli are more complex than standard motion stimuli, but it's not clear how ethological or naturalistic they really are. In particular, the occluder was the same contrast as the rest of the background, which seems like a very specific kind of occluded motion, and it's not clear how this would generalize when the occlude is the same or opposite contrast of the moving edge. Moreover, the existence of directed motion in these stimuli lead the authors to emphasize the motion on the 'preferred side', rather than just non-directional contrast changes, which seem as though they would also induce responses.

4) The modeling/decoding aspect of this paper seems pretty speculative. It doesn't seem as though these cells are known to be involved in any kind of position encoding. The fact that they transmit information about contrast changes means they can enhance position-decoding, but many other RGCs could also (better?) serve this purpose. The optic-flow-field arrangement of these cells in the retina suggests just the opposite – that they appear likely to be used for optic flow detection, in which positional information is less relevant than the field structure.

5) Last, I kept wondering how this offset excitatory input made the DSGCs look very similar to a classical Barlow-Levick model (though with DS inhibition). I believe a classical BL model would have many of the properties shown here, including the sensitivity to occluded ND motion on its 'preferred side'. Is there an advantage in the BL model formulation to having disjoint excitatory and inhibitory spatial inputs, rather than a broad excitatory field that overlaps with the delayed inhibition? If so, would such an advantage explain why this asymmetry might exist in these DSGCs, even with DS inhibition from the SACs? I guess I'm asking whether there is an advantage for general motion detection, rather than proposing a new role for these cells in localizing specific types of motion stimuli.

*Reviewer #3:*

This very interesting manuscript further describes the receptive field structure of ON-OFF retinal direction selective ganglion cells. The authors demonstrate that spot light stimuli flashed at positions that do not correspond with dendritic processes of the recorded DSGC evoke strong excitatory responses that are most powerful on the preferred side of the (moving bar determined) receptive field. The authors go onto show that small light stimuli flashed in the dendritically sampled area of visual space are also non-uniform, and maximal on the preferred side. The authors data are in line with previous reports of a nondirectional zone at the periphery of the dendritic tree of DSGCs. The experimental approaches taken by the authors seem sound. I was concerned by the obviously different kinetics of the flash response recorded under control and GABA_A_/nAChR antagonists in Figure 1 D, is this a consistent finding, what are the authors thoughts on the unusual shape of the current in Figure 1 D (lower, red trace)? As indicated in the discussion the authors have not investigated the mechanisms underlying this asymmetry, other than dismissing structural determinants (dendritic tree asymmetry, investigation of existing EM volume). This to my mind is a vital component missing from the manuscript. The authors however do go onto describe using elegant light stimulus patterns and modelling some of the potential emergent properties of this behaviour. In this reviewers mind, I am left puzzled and wanting to understand the cellular basis of the behaviour the authors have identified.

[Editors’ note: further revisions were suggested prior to acceptance, as described below.]

Thank you for resubmitting your work entitled "Spatially displaced excitation contributes to the encoding of interrupted motion by the retinal direction-selective circuit" for further consideration by *eLife*. Your revised article has been reviewed by 3 reviewers, one of whom is a member of our Board of Reviewing Editors, and the evaluation has been overseen by Ronald Calabrese as the Senior Editor.

The reviewers appreciate the revisions to the modeling work, and all agreed that those strengthened the connection to function. Several issues remain that could be further clarified or expanded. (1) The impact of the non-DS region of the RF on encoding of motion direction (Figure 7D) is mentioned briefly in the text, but deserves a more complete treatment, especially as the effects seem substantial. (2) It is not clear why the lack of asymmetry of the dendrites along the preferred-null direction is emphasized and the bias in the direction of maximum excitatory input is relegated to the supplemental figure. The latter result appears to provide at least a partial mechanistic explanation for the bias in excitatory input, and it is not clear why it is not more prominent.

*Reviewer #1:*

This paper investigates the encoding of motion stimuli by On-Off directionally-selective ganglion cells. The paper introduces a new aspect of such motion sensitivity: the lack of directional selectivity in one part of the receptive field. Corresponding to this functional asymmetry, the glutamatergic excitatory input that the cells receive is not centered on the some, but instead is displaced towards the non-directional part of the receptive field. Modeling shows that this non-directional signal could help identify the position of a moving object, particularly as it emerges from occlusion. The work in the paper is new and interesting. The paper has improved considerable in revision – particularly the new modeling sections strengthen the conclusions about function quite a bit. I do not have any comments on the science itself. There are several places in which the writing could be clearer, as detailed below.

Paragraph starting on line 210. The text here I find confusing. First, you show that excitatory input shows an asymmetry that is in decent alignment with the preferred-null axis. Then you show that the dendrites do not exhibit an asymmetry along the preferred/null axis (at least as measured by branch points and dendritic length). But then Figure S2 and the text in this paragraph argues that there is a dendritic asymmetry in the direction of the largest EPSCs. I know these are not contradictory, but if you first distill each to a simple message then they appear to be. If I am representing these results correctly, some extra text to explain how they are consistent with each other would be helpful.

Lines 128-129: this is still a bit confusing – can you say something like "the preferred side is the side of the receptive field that a bar moving in the preferred direction enters first"

Lines 338-340: Another possibility here is that the kinetics of the bipolar signals do not differ substantially.

Lines 347-350: I would move "We reasoned.…" one sentence later; at present it comes in a sentence that summarizes previous findings.

Lines 406-407: Clarify that these drugs were used individually, not as a cocktail.

Lines 462-467: This paragraph seems out of place – can it go elsewhere?

Line 501: Can you add a sentence to define how the labeled-line decoder works? It would be helpful to have a little intuition here without having to go to the Methods.

Lines 513-515: some reference for what 3 degrees of visual angle means would help (e.g. give the RF sizes in degrees).

Figure 7: can you explain in the text a bit more about why you look at different baseline rates, and why the specific rates were chosen?

Figure 7: What is the "occlusion edge" trace in B? Also the text in this figure is very small and hard to read.

Lines 533-535: suggest emphasizing that simultaneous firing of two oppositely tuned DS cells is not expected for smooth motion. And simultaneous firing of two but not all four is not expected for non-motion contrast signals.

*Reviewer #2:*

In this research, Ding and colleagues present strong evidence that the excitatory input to OO DS RGCs from bipolar cells is strongly asymmetric, with strongest inputs occurring on the side opposite from the SAC inhibition. They performed careful studies to show that this was not due to spatial asymmetry in the DSGC morphology nor to ribbon synapse density. Using 'interrupted motion' stimuli, which are effectively local directional stimuli, they show that this asymmetry leads to a non-directional response on one side of the cell's RF. Last, they created a model to show that such firing patterns could be used to improve localization of edge position under the specific conditions of an edge emerging from behind an occlusion. They also show that such a set up could lead to an 'alarm response' indicating something interesting going on when local contrast changes, and may also improve the direction-selectivity of DS RGCs.

The strength of this work lies in its careful characterization of this property of ooDRGCs, and its tests of the mechanistic origins of this property. Some similar properties have been observed before, but not characterized with the detail here. The main challenge for this paper is to figure out what the functional significance of this finding is for the animal. This paper proposes a few grounded options, but they remain quite speculative.

I have a few comments, but I'd categorize them as perhaps important but not major. The authors addressed the critical points from the first review in their revised paper. I would have liked to see more about how this property affects direction-selectivity (the known property of these cells), since I suspect it's a large and important effect, but I'm satisfied with the panel presented.

The title refers to "*the* retinal ds circuit", but this paper only focuses on one of two DS RGC types, and primarily on only 1 (sometimes 2) of the four directions of that type. It seems as though these findings apply to "a retinal DS circuit", though this depends a bit on how one defines "the retinal DS circuit".

The authors have better grounded the modeling of reversed motion and 'alarm signals' in this revision. Since it is still difficult to connect mouse retina experiments to behavior, I wonder if the paper could benefit from a broader comparison in the discussion, since I believe the non-direction-selective responsiveness of motion detectors characterized here has also been found in other well-studied systems. In particular, I believe that DS V1 cells respond non-direction-selectively to local contrast changes, so that the logic of the modeling would apply there. Similarly, studies in flies have highlighted that their local motion detectors respond to local changes in contrast, and connected it to behavior in some cases (see Fisher et al., 2015; Haag et al., 2016; Gruntman et al., 2018; Wienecke et al., 2019; Salazar-Gatzimas et al., 2018; Agrochao et al., 2020), so that the modeling logic should apply there, too. I think a discussion of how this modeling and proposed function might relate to similar features in other systems would strengthen the paper.

The authors chose to focus on a null result in Figure 2, in which dendritic arbors don't correlate strongly with direction-selectivity, but did not focus on the positive result shown in Supp 2, in which dendritic arbors correlated with the glutamatergic excitation field (S2D and E). I'm a little puzzled about why this positive result didn't warrant more discussion in the main text. Supp 2F shows no correlation between the dendritic and response RSIs, but crucially, this is only in the maximal direction for the response. If one computed the RSI for each of the 4 orientations in S2B, then the correlation would surely be significant (as it is in the other analyses in D and E). In that sense, this lack of correlation specifically in the max-opp orientation seems slightly second order. I agree that this plot can be interpreted to mean that it's not all due to dendritic arbor asymmetries, but it seems clear that the dendritic arbor has some influence on the location of the offset excitation in these neurons, so it's a little surprising that doesn't merit more attention.

L 598 typo "anther"

Note: I may have missed it, but I did not see any statement about the rawest data and analysis code being made available to readers, using Dryad and Github or another method. My impression was that this was mandatory.

*Reviewer #3:*

The revised manuscript is improved, and features more sophisticated modelling analysis. My first review emphasised a missing mechanistic explanation for the asymmetrical excitatory input received by this particular class of ON-OFF-DSGC. The authors have highlighted through analysis of dendritic morphology and a previously published EM volume that a simple structural basis for this asymmetry does not exists. They have not however investigated what the origin of the asymmetrical excitatory input is, which to this reviewers mind is a central ingredient of the work. The authors have however significantly improved their modelling work. My view of the manuscript remains unchanged, and I am left with a simple question – what is the basis of the asymmetyrical excitatory input received by this class of DSGC?

My major concern remains from the first round of review and has not been addressed in resubmission.

---

## [Author Response]

[Editors’ note: the authors resubmitted a revised version of the paper for consideration. What follows is the authors’ response to the first round of review.]

Reviewer #1:This paper describes a new finding about stimulus encoding in On-Off directionally selective ganglion cells. It is well established that these cells have spatially displaced inhibitory input from starburst amacrine cells, and that the spatial offset of inhibitory input contributes to the cells' selectivity for direction of motion. The work in this paper shows that the cells also have spatially offset excitatory input, and that this input can give rise to a non-directional response. Several functional roles are suggested for the non-directional response. I felt that the evidence for the non-directional response was strong, but that the connection to visual function was too preliminary.Functional importanceThe paper emphasizes the possible functional importance of the non-directional motion signal; this is a focus of the discussion, and is highlighted in both the abstract and introduction. I found this part of the paper less complete and convincing than the experimentally-driven results. Several issues contribute to this. One is that the contribution to identifying the position of a moving object is fairly modest.

We thank the reviewer for raising this point. In the original manuscript, the receptive field of the ganglion cell is modeled only in one dimension. In the revised manuscript, we improved our model by incorporating two-dimensional receptive fields of oppositely tuned ganglion cell populations and including moving stimuli at different speeds (Figures 7A-C, lines 486-515, lines 557-562). We found a more pronounced contribution of the displaced receptive field to position estimation during the occlusion stimulus. At the lowest bar speed, the reduction in position error when accounting for the occlusion-associated null response is over 80%, and there is around a 3 degree visual angle reduction in error across speeds. These results provide stronger support that the null-direction occlusion response benefits the localization of a moving object emerging from behind an occluder.

Another is that the impact of the non-directional component on other stimulus properties – e.g. the accuracy with which motion direction is encoded – is not explored.

Thank you for your suggestion. Because interrupted motion can evoke null-direction responses from cells near the occlusion edge, there is indeed an ambiguous signal about motion direction that can be sent to higher order visual areas. We have included an additional plot illustrating the degradation of the direction estimation during occlusion trials (Figure 7D, lines 503-505, line 563). The direction estimation error increase at the time of occlusion is quite significant across speeds, around 50 degrees.

Ultimately, the non-directional response during occlusion trials presents a trade-off between directional and positional decoding. However, this trade-off occurs only during specific visual conditions when an object emerges from behind another object. We speculate that when a moving object first appears out of an occluder, the positional information of the emerging object at that moment might be more important than the direction in which it is moving.

A third is that the position of a moving object is almost certainly encoded by multiple ganglion cell types, and hence the modest improvement in position encoding in the DS cell population may make even less contribution when the entire ganglion cell population is considered.

Thank you for raising this interesting question. We agree that multiple ganglion cell types can certainly encode object position. However, given the divergent central projection patterns of mouse ganglion cell types (over 50 retinorecipient regions in the mouse brain, Martersteck et al., 2017), the position encoding by the entire ganglion cell population may not be available to a given circuit in a retinorecipient structure. On-Off DSGCs project to the superficial layer of the superior colliculus (SC) and the shell of dorsal lateral geniculate nucleus (dLGN) (Huberman et al., 2009). In the SC, directionally tuned On-Off DSGC inputs give rise to the direction selectivity of postsynaptic collicular neurons (Shi et al., 2017), indicating that these collicular neurons do not receive retinal inputs from a broad range of RGC types, but specifically from On-Off DSGCs. While the detailed synaptic organization of the SC is unclear, increasing evidences suggest that inputs from different RGC types are further processed by parallel SC channels (Reinhard et al., 2019). Therefore, the encoding of both motion direction and location by On-Off DSGCs may benefit motion processing in their specific collicular targets. Interestingly, cells in the superficial layer of SC have been implicated in initiating both defensive and prey capture behaviors (Basso et al., 2021). Therefore, the direction and spatial location of a moving object encoded by On-Off DSGCs may help the specific collicular circuit evaluate threats or inform other behaviors.

Additionally, the synchronous response of null-direction and preferred-direction spiking can potentially provide a salient alarm signal. We have created a decoder which can distinguish whether a population of DSGCs has been presented with smooth or interrupted motion (Figures 7E-G, lines 533-551, lines 563-566). The decoder performs extremely well at low levels of baseline firing, which is a characteristic of DSGCs. This indicates that the synchronous response after discontinuous motion is extremely detectable. This detectability can potentially help indicate a change in the visual environment to higher order visual areas, which can help the animal quickly attend to the site of the change. We have revised the Discussion section to incorporate the above responses (lines 613-654).

A complete investigation of coding in the ganglion cell population is clearly too much, but a more balanced and complete consideration of the benefits and drawbacks of the mechanism described would strengthen the paper considerably.

Thank you for your comment. We have further investigated the mechanism by performing multiple new experiments as well as developed a more biologically plausible model with realistic noise levels to explore the functional significance of the receptive field properties we found.

On the experimental side, we have completed several positive controls to test the efficacy of Gabazine (Figures S5E-F, Supplemental material lines 179-183), DHbE (Figure S5G, Supplemental material lines 183-186), and gap junction blockers (Figures S5J-K, Supplemental material lines 192-194). We have also shown that the null-direction response is robust for occluded motion stimuli of different contrast configurations (Figure 5E, lines 390-396). Moreover, we have included additional analyses such as a kinetic analysis of EPSCs after the administration of DHbE and gabazine (Figure S1E, Supplemental material lines 13-15), an analysis of the inhibitory input under the occluded bar stimulus (Figure S5D, Supplemental material lines 178-180), a receptive field mapping experiment to determine the extent of the receptive field displacement on the preferred side (Figure S1H, lines 520-531, Supplemental material 22-25), and an analyses of how the separation of excitatory and inhibitory receptive fields correlates with a cell’s direction selectivity (Figure S7M, lines 462-467, Supplemental material lines 262-263), and Off dendritic analyses(Figures S2C, E, F, Supplemental material lines 76-84) and Off RF property analyses (Figures S3E-F, Supplemental material lines 142-144). These results will be further described below in the point-by-point responses to reviewers’ comments.

For the computational section, we have strengthened the model with two-dimensional receptive field placement and multiple motion speeds to show how the null-direction response may benefit position estimation of an object emerging from behind an occluder. We completed new experiments to measure the spiking activity during the fullfield and occlusion stimulus (Figures S7J-L, Supplemental material lines 259-262) and used these data for the model. We show a robust decrease in the position estimation error across speeds (Figures. 7A-C, lines 486-515, lines 557-562). Additionally, we have implemented a decoder to illustrate that the null-direction response is very detectable, which could indicate that it may be a useful signal for downstream visual areas (Figures. 7E-G, lines 533-551, lines 563-566).

Reviewer #2:[…] The work showing the asymmetry appeared careful, thorough, and well-done. The second half of the paper dealing with the functional consequences of this asymmetry left me with a few questions:1) Throughout the paper, several experiments showed no changes when a mix of receptor antagonists was added to exclude SAC inhibition as the origin of these effects. But I did not find a positive control, showing that these antagonists had the desired effect. Later, in Figures 5CD, the remaining effect after application of these antagonists was cited as evidence that the excitational asymmetry was responsible for the effect; that interpretation is only valid if the drugs truly kill all SAC input to the DSGC. What if the drugs were not 100% effective? Relatedly, in the experiments in 5CD, the measured responses all decrease with the antagonists, an effect that seems surprising and is not explained. Connecting the asymmetry in excitation to the interrupted motion is central to this paper, so it should have strong support.

Thank you for your comments, and we agree that there needs to be strong support of the efficacy of the antagonists. We have performed additional experiments and analyses as positive controls to address this concern.

The efficacy of gabazine on blocking inhibitory signaling is shown by the following results: 1. The addition of gabazine caused the spiking of DSGCs to become non-DS in response to a full-field moving bar (Figure S5E, Supplemental material lines 179182). 2. We have added new positive control experiments showing that the use of gabazine completely ablated IPSCs of DSGCs (Figure S5F, Supplemental material lines 182-183) in response to a full-field moving bar. These experiments demonstrate the efficacy of gabazine in ablating SAC inhibition and support our findings of an inhibition-independent mechanism for the displaced excitatory receptive field.

The efficacy of DHbE on blocking nicotinic excitation is verified by a new positive control experiment showing diminished EPSCs of DSGCs after DHbB application during the presentation of a full-field moving bar (Figure S5G, Supplemental material lines 183186). This experiment shows that nicotinic inputs from SACs significantly contribute to the excitatory drive of DSGCs, a finding consistent with previous reports (Lee and Zhou, 2010; Sethuramanujam et al., 2016).

Regarding the decrease of responses in Figures 5C and 5D with both gabazine and

DHbE, the decrease in EPSC is due to DHbE blocking excitatory cholinergic inputs

(Figure 5D, 386-390). For decreased spiking activity during occluded bar stimulus (Figure 5C, 383-386), we reasoned that since nicotinic excitation is a major excitatory drive of DSGC firing during visual stimulation at the RF periphery, blocking inhibition of DSGCs by gabazine is not sufficient to offset the effect of DHbE on spike reduction.

2) The measured functional results appear quite similar to results in Kuhn and Gollisch 2019, which is not cited in that context. That paper found that DSGCs responded to local contrast, not just motion, much like the results here, and suggested that oppositely tuned cells could be subtracted to eliminate this contaminating contrast signal or added to isolate the contrast signal. Here, the authors suggest a very similar use for these signals, albeit with a decoder of position and a focus on motion rather than contrast changes. (See line 528, where the authors suggest that this position-direction hypothesis is new. See also line 537: or could not be salient, if there's any kind of downstream opponent subtraction, as in primate MT.)

Thank you for this comment. We apologize for the oversight and we have added more details about the similarity between the findings of the Kuhn and Gollisch paper and our results. We have amended the discussion to discuss how both studies find that multiple subtypes of DSGCs are necessary to disambiguate multiplexed signals within the DSGC population (lines 604-611).

3) The interrupted motion stimuli are more complex than standard motion stimuli, but it's not clear how ethological or naturalistic they really are. In particular, the occluder was the same contrast as the rest of the background, which seems like a very specific kind of occluded motion, and it's not clear how this would generalize when the occlude is the same or opposite contrast of the moving edge.

We agree that showing a generalization with stimuli of different contrasts was necessary. In response to this comment, we have conducted experiments using 4 new stimuli with different contrast configurations among the occluder, the moving bar and the background (Figure 5E, lines 390-396). We show that both the On and Off responses to the new stimuli also show the occlusion null-direction response, which supports a generalization of this phenomenon across contrasts.

Moreover, the existence of directed motion in these stimuli lead the authors to emphasize the motion on the 'preferred side', rather than just non-directional contrast changes, which seem as though they would also induce responses.

Indeed, stationary flashing spots presented to the preferred side evoke responses (Figure 1). We have also added a receptive field mapping experiment to show the extent of the receptive field displacement to the preferred side (Figure S1H, Supplemental material lines 22-25). The diameter of the receptive field is 220 microns, which is larger than the average dendritic span of a DSGC.

To follow up the reviewer’s comment, we investigated whether the displaced receptive field would benefit the detection of non-directional contrast changes like a small flashing spot. We found that there is no difference in the position estimation of local contrast changes in populations with displaced receptive fields compared to populations with symmetric receptive fields in relation to the soma. Only when we decrease the size of the overall receptive field do we get an improvement in the position estimation of the local contrast change (lines 522-531). Therefore, the location information provided by synchronous preferred-direction and null-direction responses, which localizes the stimulus to a much narrower region, is necessary for improving the location estimation of the moving stimulus.

4) The modeling/decoding aspect of this paper seems pretty speculative. It doesn't seem as though these cells are known to be involved in any kind of position encoding. The fact that they transmit information about contrast changes means they can enhance position-decoding, but many other RGCs could also (better?) serve this purpose. The optic-flow-field arrangement of these cells in the retina suggests just the opposite – that they appear likely to be used for optic flow detection, in which positional information is less relevant than the field structure.

We agree that many other RGCs can also encode object position. For example, certain types of small receptive field RGCs such as W3 RGCs and HD-RGCs may also be well suited to encode object location (Jacoby et al., 2017; Kim et al., 2010; Zhang et al., 2012). However, W3 RGCs are activated only in specific instances where the background is completely uniform, such as when a bird is swooping down from an open sky, whereas On-Off DSGCs can be activated in a wide range of visual environments, including environments with noisy backgrounds. HD-RGCs also have small receptive fields, and computational modeling experiments have shown that the errors in object location between On-Off DSGCs after motion interruption and HD-RGCs are very similar (Jacoby et al., 2017). However, the position information encoded by W3 and HDRGCs may be not available to the specific downstream circuits that receive On-Off DSGC inputs. Our modeling study suggests that the population response of On-Off DSGCs after a motion interruption helps On-Off RGCs achieve the same performance as other ganglion cell populations implicated in fine spatial discrimination.

In addition to position encoding, the synchronous response of null-direction and preferred-direction spiking at the site of motion interruption can potentially provide a salient alarm signal. We have created a decoder which can distinguish whether a population of DSGCs has been presented with smooth or interrupted motion (Figures 7E-G). The decoder performs extremely well at low levels of baseline firing, which is a characteristic of DSGCs. This indicates that the synchronous response after discontinuous motion is extremely detectable. This detectability may help indicate a change in the visual environment to higher order visual areas, which can help the animal quickly attend to the site of the change.

The reviewer also mentioned optic flow detection. In this regard, another type of DSGCs, the On DSGC, which has a larger receptive field and is sensitive to global motion at slower speed, is critical for optic flow detection and the optokinetic reflex. They project to the accessory optic system in the brain stem, and provide the information of the optic flow to instruct the compensatory eye movements for stabilizing the retinal image during the animal’s self-movement. In contrast, On-Off DSGCs are considered local motion detectors since they have smaller RFs than On DSGCs, and are more sensitive to local motion compared to global motion. On-Off DSGCs project to the superficial layer of the superior colliculus (SC) and confer direction selectivity to their postsynaptic collicular neurons. SC is “a system integral for encoding spatial locations and transforming them into stimulus-directed orienting and approach behaviors” (Basso et al., 2021). The encoding of the spatial location of an emerging moving object by On-Off DSGCs may benefit rapid sensorimotor decisions that involve collicular circuitry.

As stated in the above discussions and in our revised manuscript, our conclusions about the functional significance of the null-response after motion interruption is indeed speculative, as we do not know what types of information from the retina are used by each retinorecipient brain circuit. Therefore, we resort to the modeling/theoretical approach and show that it is possible, and perhaps beneficial, that higher visual areas receive information about object position from multiple, independent channels. We wanted to show that the information of spatial location is contained within the On-Off DSGC population response, and that it is possible that higher visual centers stand to benefit from this information. We have expanded the Discussion section in the revised manuscript to include the above discussion.

5) Last, I kept wondering how this offset excitatory input made the DSGCs look very similar to a classical Barlow-Levick model (though with DS inhibition). I believe a classical BL model would have many of the properties shown here, including the sensitivity to occluded ND motion on its 'preferred side'. Is there an advantage in the BL model formulation to having disjoint excitatory and inhibitory spatial inputs, rather than a broad excitatory field that overlaps with the delayed inhibition? If so, would such an advantage explain why this asymmetry might exist in these DSGCs, even with DS inhibition from the SACs? I guess I'm asking whether there is an advantage for general motion detection, rather than proposing a new role for these cells in localizing specific types of motion stimuli.

Thank you for your comment. We have investigated whether the displacement of the excitatory and inhibitory receptive fields contributes to the directional tuning of the On-Off DSGC. Our hypothesis is that a more displaced excitatory RF to the preferred side would further minimize the impact of the weak and delayed inhibition during preferred direction motion and thereby improve direction selectivity. We checked for any correlation between the degree of displacement and the cell’s direction selectivity index (DSI) (Figure S7M, Supplemental material lines 262-263). To quantify the displacement between excitation and inhibition, we determined the onset times of the EPSC and the IPSC in the preferred direction and subtracted the EPSC onset time from the IPSC onset time. If the difference is positive, that means that the IPSC occurs later than the EPSC. We have shown that the cells with an earlier onset in EPSC compared with IPSC in the preferred direction have higher DSIs. This means that the cells with more displaced excitatory RFs compared with inhibitory RFs are generally more direction selective, indicating a potential benefit of the displacement on the cell’s directional tuning (lines 462-467). We have included this analysis in the revised manuscript.

Reviewer #3:This very interesting manuscript further describes the receptive field structure of ON-OFF retinal direction selective ganglion cells. The authors demonstrate that spot light stimuli flashed at positions that do not correspond with dendritic processes of the recorded DSGC evoke strong excitatory responses that are most powerful on the preferred side of the (moving bar determined) receptive field. The authors go onto show that small light stimuli flashed in the dendritically sampled area of visual space are also non-uniform, and maximal on the preferred side. The authors data are in line with previous reports of a nondirectional zone at the periphery of the dendritic tree of DSGCs. The experimental approaches taken by the authors seem sound. I was concerned by the obviously different kinetics of the flash response recorded under control and GABA_A_/nAChR antagonists in Figure 1 D, is this a consistent finding, what are the authors thoughts on the unusual shape of the current in Figure 1 D (lower, red trace)?

Thank you for this comment about kinetics. We have added a new analysis of the kinetics of the flash response (Figure S1E, lines 171-173, Supplemental material lines 13-15). We have found that the addition of the GABA_A_/nAChR antagonists decrease the rise time and the decay time of the response. We reason that the thinner and faster current is due to the blockage of the cholinergic component of the EPSC by the nAChR antagonist. When bipolar cells are activated, they activate both the DSGC and the starburst amacrine cell at the same time. Therefore, the total EPSC of the DSGC is the integration of the monosynaptic glutamatergic input from bipolar cells and the slower disynaptic cholinergic input from the starburst amacrine cells. Removing the cholinergic component of the EPSC by DHbE thereby isolates the more transient waveform of the glutamatergic component.

As indicated in the discussion the authors have not investigated the mechanisms underlying this asymmetry, other than dismissing structural determinants (dendritic tree asymmetry, investigation of existing EM volume). This to my mind is a vital component missing from the manuscript. The authors however do go onto describe using elegant light stimulus patterns and modelling some of the potential emergent properties of this behaviour. In this reviewers mind, I am left puzzled and wanting to understand the cellular basis of the behaviour the authors have identified.

Thank you for your comment. We share the same interest as the reviewer on identifying underlying cellular mechanisms of the excitatory RF displacement. We have been able to exclude several mechanisms and pinpoint what kind of input must be creating the asymmetry. While we have not narrowed this down completely, we feel that we have made several significant advances not only in characterizing this phenomenon, but also in guiding future work towards its mechanistic origin.

In this paper, we have identified a spatially displaced excitatory receptive field that emerges upon partial receptive field stimulation. We have shown that the excitatory receptive field is independent of inhibitory input from starburst amacrine cells, and have identified that the excitatory asymmetry is due to glutamatergic bipolar input. In our revised manuscript, we have added additional positive controls to verify the efficacy of the GABAergic and cholinergic antagonists. We have also performed new experiments at different stimulus contrasts to show that the displaced receptive field is evident under different visual conditions.

Using EM microscopy and dendritic reconstruction, we have ruled out bipolar input distribution and gross dendritic morphology for the mechanism underlying the displaced excitatory receptive field. With a receptive field mapping experiment, we have shown that there is a non-uniform glutamatergic conductance across the dendritic span of On-Off DSGCs. We agree that it would be fascinating to uncover a detailed mechanism of the displaced excitatory receptive field. We are planning to investigate dendritic and synaptic mechanisms in a continuation of this project. In the future, we plan on evaluating the density of glutamate receptors along the dendritic span as well as the dendritic integration properties of the On-Off DSGCs.

Having identified an interesting receptive field property, we were also wondering what potential functional significance the property may play in visual processing. Using a population vector decoder, we show that the null-direction response arising from the displaced RF during partial RF activation can help the population distinguish object location. This suggests that On-Off DSGCs can encode multiple visual features, of motion direction and of object location. Though we have not completely elucidated the mechanism, we hope to share our current results with our colleagues in a timely manner. We believe that our study will be of interest to retinal physiologists as well as computational neuroscientists by investigating how a previously overlooked cellular property can benefit population coding during visual processing.

[Editors’ note: what follows is the authors’ response to the second round of review.]

The reviewers appreciate the revisions to the modeling work, and all agreed that those strengthened the connection to function. Several issues remain that could be further clarified or expanded.(1) The impact of the non-DS region of the RF on encoding of motion direction (Figure 7D) is mentioned briefly in the text, but deserves a more complete treatment, especially as the effects seem substantial.

We thank the reviewers for the comment and have now expanded on the impact of the non-DS RF region on the encoding of motion direction. We include a new figure panel (revised Figure 7 C) to plot the direction encoding error over time. Together with Figure 7B, they demonstrate that when an object emerges from an occluder, the DSGC population response transiently prioritizes the encoding of the emerging moving object position over its motion direction. We have also revised the text accordingly to provide more discussion about the relationship between position and direction encoding during the occluded motion stimulus.

(2) It is not clear why the lack of asymmetry of the dendrites along the preferred-null direction is emphasized and the bias in the direction of maximum excitatory input is relegated to the supplemental figure. The latter result appears to provide at least a partial mechanistic explanation for the bias in excitatory input, and it is not clear why it is not more prominent.

Thank you for raising this point. We agree with the reviewers that the dendritic morphology partially contributes to the shape of the excitatory RF. Previously we put this point in the supplemental figure since we thought this is not very surprising given that the dendrites provide the physical substrate for the excitatory synapses. We have now followed the reviewers’ suggestions to move this information to the main figure (Figure 2E).

We also add a new main figure panel (Figure 2F) from the previous supplemental figure to illustrate that the spatial skew of the excitatory RF map cannot be solely explained by the dendritic skew, because there is no positive correlation between the extent of the dendritic skew and the extent of the EPSC skew (Figure 2F). Together, these results show that the spatial organization of excitatory RF can be influenced, but cannot be solely explained, by dendritic arbor distribution. We have also revised the text to clarify this conclusion.

Reviewer #1:[…] The work in the paper is new and interesting. The paper has improved considerable in revision – particularly the new modeling sections strengthen the conclusions about function quite a bit. I do not have any comments on the science itself. There are several places in which the writing could be clearer, as detailed below.Paragraph starting on line 210. The text here I find confusing. First, you show that excitatory input shows an asymmetry that is in decent alignment with the preferred-null axis. Then you show that the dendrites do not exhibit an asymmetry along the preferred/null axis (at least as measured by branch points and dendritic length). But then Figure S2 and the text in this paragraph argues that there is a dendritic asymmetry in the direction of the largest EPSCs. I know these are not contradictory, but if you first distill each to a simple message then they appear to be. If I am representing these results correctly, some extra text to explain how they are consistent with each other would be helpful.

We appreciate this feedback and have revised this part of the text and Figure 2 to better explain the relationship between dendritic morphology and EPSC displacement. In the revised manuscript, we use the following order in the text (lines 178-196):

1. EPSCs are skewed toward the preferred side (Figure 1).

2. Dendritic fields are not consistently skewed toward the preferred side (Figures 2A-2D).

3. Dendritic fields partially influence ESPC displacement (Figure 2E)

4. Dendritic fields cannot solely explain EPSC displacement (Figure 2F), suggesting additional mechanisms also playing a role.

Lines 128-129: this is still a bit confusing – can you say something like "the preferred side is the side of the receptive field that a bar moving in the preferred direction enters first".

Thank you for your suggestion. We have amended the sentence to include your suggestion (lines 128-130).

Lines 338-340: Another possibility here is that the kinetics of the bipolar signals do not differ substantially.

We have amended the sentence to include this possibility (lines 271-273).

Lines 347-350: I would move "We reasoned.…" one sentence later; at present it comes in a sentence that summarizes previous findings.

Thank you for your suggestion. We have moved the sentence later (lines 289-292).

Lines 406-407: Clarify that these drugs were used individually, not as a cocktail.

We have clarified that these drugs are used individually (lines 317-319).

Lines 462-467: This paragraph seems out of place – can it go elsewhere?

Thank you for pointing this out. We have moved the paragraph earlier to lines (278-285)

Line 501: Can you add a sentence to define how the labeled-line decoder works? It would be helpful to have a little intuition here without having to go to the Methods.

We have included a brief description of the labeled-line decoder in the Results section (line 396-399):

“We used a labeled-line decoder (equivalent to a “population vector” decoder) that estimated the spatial position of the moving bar edge as a weighted average of the RF center positions where the weights were determined by the firing rate (response strength) and RF width (response precision) of each cell (see Methods).”

Lines 513-515: some reference for what 3 degrees of visual angle means would help (e.g. give the RF sizes in degrees).

We have amended the text according to the suggestion:

“The absolute position error decrease was around 3 degrees of visual angle, or around half the receptive field size of an On-OFF DSGC, across bar speeds in models with low levels of background noise.” – amended sentence (line 421)

Figure 7: can you explain in the text a bit more about why you look at different baseline rates, and why the specific rates were chosen?

Thank you for your suggestion. We have added more information about why we chose to look at different baseline rates (lines 402-406). We wanted to match the range of baseline rates reported in the literature.

“We chose to implement the model with different low baseline firing rates to more faithfully represent biological noise. Previous reports show baseline firing rates up to 0.1 Hz (Yao et al., 2018) for On-Off DSGCs. Our experiments yielded background noise levels more on the scale of 0 – 0.025 Hz. Thus, we evaluated the computational model at three different low noise levels.” – amended sentence

Figure 7: What is the "occlusion edge" trace in B? Also the text in this figure is very small and hard to read.

Thank you for noting these issues. We have revised Figure 7B. Instead of using a line, we use a shaded area indicating the position of the occluder. We have also increased the font size of this figure for easier reading.

Lines 533-535: suggest emphasizing that simultaneous firing of two oppositely tuned DS cells is not expected for smooth motion. And simultaneous firing of two but not all four is not expected for non-motion contrast signals.

Thank you for this excellent suggestion. We have included the following text (lines 445-450):

“During smooth motion, only one subtype of DS cell would respond to a bar moving across the visual scene. However, synchronous firing of two oppositely tuned DS cells would occur to represent interrupted or emergent motion. The synchronous firing of two DSGC subtypes can be a unique signature of encoding interrupted motion, which can differ from the encoding of non-motion contrast signals where the recruitment of all four DSGC subtypes would be expected.”

Reviewer #2:[…] The strength of this work lies in its careful characterization of this property of ooDRGCs, and its tests of the mechanistic origins of this property. Some similar properties have been observed before, but not characterized with the detail here. The main challenge for this paper is to figure out what the functional significance of this finding is for the animal. This paper proposes a few grounded options, but they remain quite speculative.I have a few comments, but I'd categorize them as perhaps important but not major. The authors addressed the critical points from the first review in their revised paper. I would have liked to see more about how this property affects direction-selectivity (the known property of these cells), since I suspect it's a large and important effect, but I'm satisfied with the panel presented.The title refers to "the retinal ds circuit", but this paper only focuses on one of two DS RGC types, and primarily on only 1 (sometimes 2) of the four directions of that type. It seems as though these findings apply to "a retinal DS circuit", though this depends a bit on how one defines "the retinal DS circuit".

Thank you for pointing this out. We have changed the title to “a retinal ds circuit”. We also modified the abstract to indicate that we are referring to On-Off DSGCs tuned to motion in the horizontal motion axes.

The authors have better grounded the modeling of reversed motion and 'alarm signals' in this revision. Since it is still difficult to connect mouse retina experiments to behavior, I wonder if the paper could benefit from a broader comparison in the discussion, since I believe the non-direction-selective responsiveness of motion detectors characterized here has also been found in other well-studied systems. In particular, I believe that DS V1 cells respond non-direction-selectively to local contrast changes, so that the logic of the modeling would apply there. Similarly, studies in flies have highlighted that their local motion detectors respond to local changes in contrast, and connected it to behavior in some cases (see Fisher et al., 2015; Haag et al., 2016; Gruntman et al., 2018; Wienecke et al., 2019; Salazar-Gatzimas et al., 2018; Agrochao et al., 2020), so that the modeling logic should apply there, too. I think a discussion of how this modeling and proposed function might relate to similar features in other systems would strengthen the paper.

Thank you for the helpful suggestion. We have included an additional paragraph discussing the interesting parallels of local contrast responses in vertebrate and fly motion detectors (lines 516-530).

“Elements of the visual processing scheme implicated in our study parallel those in other visual areas and species. […] Interestingly, local contrast response properties of direction-selective neurons in the fly visual system have been shown to profoundly modulate their motion computations and contributes to visually guided behavior (Clark et al., 2014; Drews et al., 2020; Matulis et al., 2020).”

The authors chose to focus on a null result in Figure 2, in which dendritic arbors don't correlate strongly with direction-selectivity, but did not focus on the positive result shown in Supp 2, in which dendritic arbors correlated with the glutamatergic excitation field (S2D and E). I'm a little puzzled about why this positive result didn't warrant more discussion in the main text. Supp 2F shows no correlation between the dendritic and response RSIs, but crucially, this is only in the maximal direction for the response. If one computed the RSI for each of the 4 orientations in S2B, then the correlation would surely be significant (as it is in the other analyses in D and E). In that sense, this lack of correlation specifically in the max-opp orientation seems slightly second order. I agree that this plot can be interpreted to mean that it's not all due to dendritic arbor asymmetries, but it seems clear that the dendritic arbor has some influence on the location of the offset excitation in these neurons, so it's a little surprising that doesn't merit more attention.

Thank you for your comments. A related question is raised by reviewer 1 and in the summary statement. We agree with the reviewers that the dendritic morphology partially contributes to the shape of the excitatory RF. Previously we put this point in the supplemental figure since we thought this is not the most important finding given that the dendrites provide the physical substrate for the excitatory synapses. We have now followed the reviewers’ suggestions to move this information to the main figure (Figure 2E). We also add a new main figure panel (Figure 2F) from the previous supplemental figure to illustrate that the spatial skew of the excitatory RF map cannot be solely explained by the dendritic skew, because there is no positive correlation between the extent of the dendritic skew and the extent of the EPSC skew (Figure 2F). Together, these results show that the spatial organization of excitatory RF can be influenced, but cannot be solely explained, by dendritic arbor distribution. We have revised the text to clarify this conclusion. (lines 178-196).

L 598 typo "anther".

Thank you for this correction. We have fixed this typo.

Note: I may have missed it, but I did not see any statement about the rawest data and analysis code being made available to readers, using Dryad and Github or another method. My impression was that this was mandatory.

We have uploaded a statement on the transparency form that the raw data and analysis code will be available on the *eLife* website, Github (jnnfr-ding/Occulsion-model) and Dryad.

Reviewer #3:The revised manuscript is improved, and features more sophisticated modelling analysis. My first review emphasised a missing mechanistic explanation for the asymmetrical excitatory input received by this particular class of ON-OFF-DSGC. The authors have highlighted through analysis of dendritic morphology and a previously published EM volume that a simple structural basis for this asymmetry does not exists. They have not however investigated what the origin of the asymmetrical excitatory input is, which to this reviewers mind is a central ingredient of the work. The authors have however significantly improved their modelling work. My view of the manuscript remains unchanged, and I am left with a simple question – what is the basis of the asymmetrical excitatory input received by this class of DSGC?My major concern remains from the first round of review and has not been addressed in resubmission.

We understand the reviewer’s concern and are currently working on additional experiments regarding the mechanism. We share our preliminary plots with the reviewers in the attachment. However, we think that this result is too preliminary to include in this manuscript, and plan to corroborate it with additional functional and anatomical characterizations for more definitive and detailed mechanistic insights.